# Morning boundary layer conditions for shallow to deep convective cloud evolution during the dry season in the Central Amazon

Alice Henkes[1,4], Gilberto Fisch[1,2], Luiz Augusto Toledo Machado[1,3], and Jean-Pierre Chaboureau[4]

[1]National Institute for Space Research, Cachoeira Paulista, São Paulo, Brazil.
[2]Universidade de Taubaté (UNITAU), Taubaté – São Paulo, Brazil
[3]Multiphase Chemistry Department, Max Planck Institute for Chemistry, Mainz, Germany
[4]Laboratoire d´Aérologie, Université de Toulouse, CNRS, UPS, Toulouse, France

**Correspondence:** Alice Henkes (alice.henkes@inpe.br)

**Abstract.** Observations of the boundary layer (BL) processes are analyzed statistically for dry seasons of two years and in detail, as case studies, for four shallow convective days (ShCu) and four shallow-to-deep convective days (ShDeep) using a suite of ground-based measurements from the Observation and Modeling of the Green Ocean Amazon (GoAmazon 2014/5) Experiment. The BL stages in ShDeep days, from the nighttime to the cloudy mixing layer stage, are then described in compar­

ison with ShCu days. Atmospheric thermodynamics and dynamics, environmental profiles, and surface turbulent fluxes were employed to compare these two distinct situations for each stage of the BL evolution. Particular attention is given to the morn­ing transition stage, in which the BL changes from stable to unstable conditions in the early morning hours. Results show that the decrease in time duration of the morning transition on ShDeep days is associated with high humidity and well established vertical wind shear patterns. Higher humidity since nighttime not only contributes to lowering the cloud base during the rapid

growth of the BL but also contributes to the balance between radiative cooling and turbulent mixing during nighttime, resulting in higher sensible heat flux in the early morning. The sensible heat flux promotes rapid growth of the well-mixed layer, thus favoring the deeper BL starting from around 08:00 LST (UTC-4 h). Under these conditions, the time duration of morning transition is used to promote convection having an important effect on the convective BL strength leading to the formation of shallow cumulus clouds and their subsequent evolution into deep convective clouds. Statistical analysis was used to validate the

conceptual model obtained from the case studies. Despite the case-to-case variability, the statistical analyses of the processes in the BL show that the described processes are well representative of cloud evolution during the dry season.

## 1 Introduction

Over land under clear sky and in the absence of any synoptic disturbance, the atmospheric boundary layer (BL) presents a structure that evolves according to a daily cycle of a stable and weakly turbulent at night and an unstable and strongly turbulent

during the day (Stull, 1988). The diurnal cycle of the BL processes exerts a strong control on the initiation of convection, especially when mid-and upper-tropospheric forcing is weak (Wilson et al., 1998). BL processes are frequently separated into surface effects, BL effects, and wind circulations (Pielke, 2001). Tightly coupled to the underlying land surface by a strong diurnal cycle of surface fluxes, shallow cumulus clouds can form at the top of the BL. Although generally associated with

favorable weather conditions, several other factors can potentially dictate whether shallow convection will develop into deep convection, such as free atmospheric moisture entrainment, vertical wind shear, cloud aerosol interactions (Chakraborty et al., 2018), and local land-atmosphere interactions (Fitzjarrald et al., 2008).

The role of BL processes and the overlying atmosphere in cloud formation and its transition from shallow to deep clouds remains an important and complex issue in both observational (Zhang and Klein, 2010; Ghate and Kollias, 2016; Tang et al., 2016; Zhuang et al., 2017; Chakraborty et al., 2018; Schiro and Neelin, 2018; Biscaro et al., 2021) and modeling studies where high-resolution models have been used to provide information on the variables that control shallow and deep convection (Chaboureau et al., 2004; Khairoutdinov and Randall, 2006; Wu et al., 2009). Examining the BL evolution from nighttime to daytime has important implications for understanding convection, cloud formation, and the exchange of chemical species between land surface and atmosphere . Indeed, the onset time of a fully convective BL is determined by the surface heat fluxes and the time necessary to eliminate the inversion of the stable BL after sunrise in the morning (erosion of the nocturnal BL). The timing of the increase in convective turbulence in the morning transition is essential for the convection to penetrate throughout the BL, transport aerosol, and then form shallow clouds in the upper BL. This is particularly true over Amazonia rainforest (Martin et al., 1988). For example, in the morning, particles or atmospheric trace gases produced in the forest are transported from the surface to the troposphere. In the afternoon, they are removed by precipitation scavenging (Machado et al., 2021).

Deep convective clouds are a ubiquitous feature of the atmospheric environment of the Amazonia rainforest (Oliveira et al., 2020). Therefore, during the years 2014 and 2015 in the Central Amazon region, the Observations and Modeling of the Green Ocean Amazon (GoAmazon2014/5 Experiment; Martin et al., 2017) project was carried out to assess the complex interaction between the plume of pollution generated in the city of Manaus-Amazonas, the clouds dynamics, and the underlying vegetation (forest and pasture). One of the goals of the project related to the cloud life cycle was to understand physical processes on daily transition in cloud development from shallow to deep convection during wet and dry seasons. Zhuang et al. (2017) using the GoAmazon 2014/5 observations, showed the different environmental characteristics and seasonal variations of the transition from shallow to deep convection during the two years of the campaign. The authors found that during the dry season, from June to September, a deep moist layer extending from the BL to the free troposphere is required to transition from shallow to deep convection. Comparing the days with the locally formed daytime deep convective clouds (transition case) to those with only shallow and congestus convective clouds (non-transition), they suggested that in the early morning, between 07:00–09:00 Local Standard Time (LST, the local standard time in Amazonia is UTC-4), the 0–3 km large vertical wind shear (VWS) may appear to link shallow-to-deep cloud conditions during the dry season in contrast to the wet and dry-to-wet transition seasons.

The dry season in the Amazonia is frequently characterized by shallow convective clouds and on some days, there is locally deep moist convection in the early afternoon (Ghate and Kollias, 2016; Tang et al., 2016; Giangrande et al., 2020). The convective regime has a typically bimodal cloud top distribution (Wang et al., 2018), an absence of mid-level clouds, and a shallow-to-deep transition signature (Giangrande et al., 2020). Giangrande et al. (2017) provided a composite analysis of the cloud type frequency collected during the 2-year of GoAmazon campaign. They found that shallow cumulus clouds and deep convective clouds represent 22.1% and 5.2% of the total cloud frequencies of cloud type occurrence year round, respectively. While for the dry season, the numbers are reduced to 16.6% and 1.5%, respectively. Despite the lower cloud frequency, the rain

rate caused by moist deep convection during the dry season is typically more intense than that of the wet season (Zhuang et al., 2017; Machado et al., 2018). Tian et al. (2021) recently expanded the previous convective regime classification of Zhang and Klein (2010, 2013) based at the U.S. Southern Great Plains (SGP) site to the GoAmazon main site. Their criteria associated days with local deep convective clouds to a presence or not of .

Although previous studies, such as those by Ghate and Kollias (2016); Tang et al. (2016); Zhuang et al. (2017); Chakraborty et al. (2018); Tian et al. (2021) provide observational evidences and explanations for understanding the physical processes controlling the transition from shallow to deep convective clouds, some information is still missing regarding the role of water vapor and wind shear in the erosion of the nocturnal BL and the evolution to deep convection. The dry season is predominantly associated with large-scale subsidence that dries out the troposphere and suppresses large-scale deep convective clouds. However, it has been demonstrated that the probability of locally occurring daytime deep convective clouds, in the drier months (e.g. June, July, August and September), is strongly tied to both the lower-free tropospheric and the BL moisture (e.g., Schiro and Neelin, 2018). During the diurnal cycle of convective BL in the dry season, the enhanced shallow and deep convection are related to increased humidity in the lower free troposphere (Ghate and Kollias, 2016; Giangrande et al., 2020) . This study examines the BL processes that influence convective cloud development, particularly during the dry season, when the BL contributes more to the total column moisture than during wet season (Schiro and Neelin, 2018).

Martin et al. (1988) studying the dry season during the NASA Amazon Boundary Layer Experiment (GTE ABLE 2B), described the structure and evolution of the convective mixing layer for the days without convective activity (e.g., common undisturbed conditions). The present study extends the BL study for shallow and shallow-to-deep convective regimes by examining the Amazonian BL transition from night to day. Thus, for the Amazonian BL, we adopt the previous definition of mixed layer growth of Stull (1988) to a cloudy BL evolution from stable to unstable conditions in the dry season. The subject of this study considers the evolution of four BL stages , as follows: (1) the nocturnal and stable BL stage, (2) the morning transition stage, (3) the rapid growth of the convective BL stage, and (4) the cloudy mixed layer stage. The definition of the BL stages matches with previous studies about morning transition and erosion of nocturnal BL (Angevine et al., 2001; Carneiro and Fisch, 2020), growth of BL (Tennekes, 1973; Stull, 1988; Martin et al., 1988) and Amazonian cloudy mixing layer (Betts and Viterbo, 2005).

This paper aims to understand the dominant processes in the shallow-to-deep transition during the dry season and to explore the physical mechanisms of the convective cloud-BL interaction. It examines the evolution of the BL structure between nighttime and the fully developed convective stage during typical shallow cumulus days and typical shallow-to-deep transition days. For this, we use ground-based observation data at the T3 site (the GoAmazon main site) (Martin et al., 2017) of GoAmazon 2014/5. The results are organized by detailed case studies of shallow and deep convective clouds days, the development of a conceptual model and a statistical analysis to verify the adherence of the conceptual model to a larger set of cases during the two dry season years.

The article is organized as follows. Section 2 presents the data and the methodology used to identify the cloudy BL stages and to select the convective days. Section 3, based on case studies, is primarily descriptive of the environmental conditions and atmospheric stability in which the BL grows that are observed by radiosondes, the surface effects in terms of the daily

evolution of meteorological variables at near-surface level, and the atmospheric BL effects on shallow and shallow-to-deep

cases. Section 4 present the statistics of BL processes during the two dry seasons and the conclusions and next steps are presented in section 5.

## 2 Data and methodology

### 2.1 Observations at GoAmazon 2014/5 site

The T3 site was the main site of the GoAmazon 2014/5 and it is located at Manacapuru (03° 12' 36" S; 60° 36' 00" W; 50 m),

70 km downwind of Manaus, in the central part of the state of Amazonas, Brazil. The T3 was a pasture site of 2.5 km by 2 km surrounded by forest, with the forest canopy height of approximately 35m (Martin et al., 2017). During the GoAmazon 2014/5 campaign, observational data were collected by the Atmospheric Radiation Measurement (ARM) Mobile Facility (AMF-1, Mather and Voyles, 2013).

Meteorological variables at near-surface level were measured by five different instruments. The average of surface air tem-

perature, specific humidity, and wind speed are from the Surface Meteorological system at 30 min time average. Precipitation is used from the Present Weather Detector herein in the conditional rain rate (e.g., mean rain rates are determined only for the periods when rainfall is equal to or greater than 1 mm h$^{-1}$) to selected the days with precipitation. The three wind components, surface sensible and latent heat fluxes are obtained from the Eddy CORrelation Flux (ECOR) (ARM, 2014b) measurement system 3 m above the ground in 30 min averages. Soil moisture (centered at 2.5 cm of depth), and the average soil surface heat

flux from the ARM Surface Energy Balance System (ARM, 2014d) at 30 min time resolution. The surface net long-wave and shortwave radiative flux were extracted from the ARM RADFLUX (ARM, 2014c) Value Added Product in 30 min averages.

The BL height was retrieved by a laser ceilometer (ARM, 2014a) operating at 905-nm wavelength. The ceilometer retrieves three BL height candidates provided by Vaisala BL-view software at 16 s resolution. The retrieval approach determines the layers associated with the aerosol backscatter gradient profile as possible BL height candidates. Since high aerosol concentra-

tion often occurs at the base (or interior) of the entrainment zone in convective conditions or at the level of the temperature inversion capping the residual layer, in neutrally stratified conditions. The first BL candidate is associated with the BL height while the others (second and third estimates) with residual layers (Poltera et al., 2017; Carneiro and Fisch, 2020). Therefore, the first BL height candidate was used to determine the height during the diurnal evolution of the atmospheric BL in 5 min average. We refer to the BL height alternatively as the BL top or the BL depth associated with the BL mixing layer. The second

and third BL candidates were used to estimate the hourly average of residual layer height. In order to determine the cloud base, the lowest estimates from ceilometer was used. In addition, the onset of turbulence from the nocturnal BL to morning phase was determined by the vertical wind standard deviation data from the Sonic Detection and Ranging (SODAR) wind profiler (ARM, 2014e) at 30 min time resolution that measures the wind profile ranging from 30 to 400 m.

The environmental condition profile was measured with radiosondes (ARM, 2014f) launched over the T3 site at regular

intervals around 01:30, 07:30, 10:30, 13:30, and 19:30 LST during the second Intensive Operating Period (IOP2). Lifting condensation level (LCL), Convective Available Potential Energy (CAPE), and Convective Inhibition (CIN) are computed

using sounding data assuming an air parcel in which level of the maximum virtual temperature is in the first 1000 m. Due to the vertical resolution that varies with each launch and the atmospheric conditions, data were linearly interpolated with a resolution of 20 m.

The cloud type profile were extracted from the cloud type product by the ARM Active Remote Sensing of Clouds (ARSCL). The RWP-WACR-ARSCL cloud mask product (Feng and Giangrande, 2018) estimates the cloud type profile by combining data from the W-band ARM cloud radar (WACR), radar wind profiler, micropulse lidar, and ceilometer (Giangrande et al., 2017; Feng and Giangrande, 2018). Reflectivity data from the S-band radar, located in Manaus which is 67.8 km northeast of T3, was used to verify the presence of deep convective clouds in an area within 60 km of T3.

## 2.2    The cloudy boundary layer stages

The BL evolution was divided into four stages according to its daily cycle, the end of each stage corresponding to the beginning of the next stage. Each stage is defined as follows:

*(i) Stable stage*: The first stage occurs during the nighttime, defined here between 21:00 LST and sunrise (at 06:00 LST).

*(ii) Morning transition stage*: It is defined as the succession of 3 events: (1) the sunrise, (2) the time when the sensible

heat flux becomes positive defined as crossover, and (3) the onset of the convective BL, when the erosion of nocturnal BL is completed and the growth rate of the BL height reaches 100 m h$^{-1}$ (Stull, 1988). The classification of these events follows the one proposed by Angevine et al. (2001). The complete erosion of the nocturnal BL, during dry season in Amazon region, usually occurs two hours after sunrise (Carneiro et al., 2020).

*(iii) Rapid growth stage*: This stage starts 2 to 3 hours after sunrise, depending on the duration of the morning transition.

It ends when the BL height begins to decrease due to the emergence of cumulus clouds. This stage usually starts around 08:00–09:00 and ends around 11:00 LST.

*(iv) Cloudy mixing layer stage*: The last stage is between around 11:00 and 15:00 LST. It corresponds to the occurrence of a cloudy mixing layer where a deeper convective BL is observed with cumulus clouds. During this stage the transition between shallow and deep convection may occur or not.

## 2.3    Selection of shallow convective and shallow-to-deep convective case studies

Typical shallow and shallow-to-deep convective case studies days were searched in the dry season during the IOP2 from GoAmazon 2014/5 (from 15 August to 15 October 2014; Martin et al., 2016; Giangrande et al., 2017). This selection period was limited to the period from 22 August to 15 October, when an additional radiosonde was launched at 10:30 LST to enhance the diurnal radiosonde coverage. The period from 13 to 29 September was excluded from the analysis as no WACR radar data was

available. The beginning of October was characterized by very intense rainy days, with a typical wet season behavior, therefore the October days were not considered in this selection. Cloud-type definitions based on cloud boundaries and thickness are from Giangrande et al. (2017) and the criteria to characterize shallow and shallow-to-deep convective days are adapted from Zhang and Klein (2010) and from Zhuang et al. (2017). A day was classified as representative of shallow or shallow-to-deep cloud day if it simultaneously satisfied the following conditions:

*Shallow convective days (ShCu):* They are defined as days with shallow cumulus in late morning or early afternoon observed by the RWP-WACR-ARSCL product. These clouds should have cloud top height under 3 km, and the cloud base height under 2 km. In the region within a radius of 60 km of the T3 site, the reflectivity of the S-band radar should be less than 20 dBz, at all hours of the day, ensuring there is no deep convection around the T3 site. No precipitation is measured at any time of the day.

    *Shallow-to-deep convective days (ShDeep):* They are characterized by shallow cumulus clouds observed during the growth
of the convective layer stage, with cloud base within the BL (around 1-2 km depth) and rising gradually over time until convective development from shallow to deep. Unlike ShCu days, the transition from shallow-to-deep cloud cover is observed by the RWP-WACR-ARSCL product. The deep convective cloud was characterized by a cloud base height at the top of the BL (around 1-2 km), a top above 3 km, and a thickness equal or greater than 5 km. Precipitation occurs during the cloudy mixing layer stage and is higher than 1 mm h$^{-1}$.

From 22 August to 13 September 2014, the following general cases were observed: 5 cases of shallow clouds evolving to deep convection around T3 (but not above), 2 cases of deep convection during the night, 7 cases of shallow clouds to congestus clouds, and 4 cases of ShCu days and 4 cases of ShDeep days. These latter 8 cases were selected fulfilling the requirements presented above. Days of ShDeep evolve from shallow to deep convection at different times, all in the afternoon (Fig. 1) between 12:00 and 16:00 LST. The precedent nights of these cases has no deep convective clouds, but generally had few cirrus
clouds above 8 km. The dates and the list of relevant features are given in Table 1. Figure 1 presents a description of the selected days showing the cloud-type classification from the merged RWP-WACR-ARSCL product on each ShCu and ShDeep day.

## 3    Case studies and results

Based on these selected cases, the different physical processes associated with these two different patterns of daily evolution were evaluated under similar large-scale conditions during the late dry season (IOP2). Radiosonde observations at high tempo-
ral resolution, thanks to the additional radiosonde in the morning, allowed us to investigate the rapid growth of BL. Of course, different phases of the cloud life cycle and cloud amounts can alter the BL structure, so after the case studies, a statistical evaluation with a larger number of cases was performed in order to validate the main characteristics driving the shallow and shallow to deep convective days processes.

### 3.1    Environmental conditions and atmospheric stability

The soundings are used to investigate environmental conditions associated to the BL evolution.The BL evolution stages, as defined above, was evaluated using the following radiosondes: the stable stage at 1:30 LST, the morning transition stage at 7:30 LST, the rapid growth stage at 10:30 LST, and the cloudy mixing layer stage at 13:30 LST. Figure 2 shows the vertical profiles for relative humidity (RH), potential temperature, specific humidity, and wind speed measured by radiosonde on ShDeep and ShCu days. The differences between ShDeep and ShCu days are compared by the mean (red and blue lines)
and its standard deviation (shaded region) in the lowest 3500 m. Comparing the RH profiles between ShDeep and ShCu days, there are significant difference in the moisture vertical profiles (Fig. 2a). Not only is the humidity higher on ShDeep days, but

it is almost constantly distributed along the vertical direction. At all stages, the RH is 10 % higher on ShDeep days in the layer beneath 2000 m than on ShCu days. Above 2000 m, the differences between ShDeep and ShCu are even more apparent. On average, the RH on ShDeep days are 40 % higher than ShCu days during the stable stage and morning transition and up to 50 % during the rapid growth stage and cloudy mixing layer stage. This reflects the importance of mid-level humidity during the dry season as observed by Ghate and Kollias (2016); Zhuang et al. (2017), and Chakraborty et al. (2018). Our results is similar to the one presented by Zhuang et al. (2017) for shallow-to-deep convective cases during the daytime, even considering a distinct classification of shallow and deep convective cases.

The vertical distribution of water vapor is higher on ShDeep days than on ShCu days from the stable stage (01:30 LST) to the morning transition stage (07:30 LST) as well as during the daytime stages 10:30-13:30 LST (Fig. 2b). More specifically, the temporal evolution of specific moisture profiles clearly show that ShDeep days are moister (by 2.0 g kg$^{-1}$) than ShCu days. The layer above BL from stable stage to morning transition stage also differs on ShDeep and ShCu days. For instance, on the ShDeep day profiles, the specific humidity is higher by about $\sim$0.5 g kg$^{-1}$ at $\sim$ 1200 m, while on ShCu days, it is lower by about $-1.0$ g kg$^{-1}$. During the rapid growth stage, there is a difference of about 2.0 g kg$^{-1}$ under 3000 m. During the cloudy mixing layer stage, there is a difference of about 3.0 g kg$^{-1}$ beneath 1000 m, and above this level there is a decrease with height.

The potential temperature profiles are shown in Fig. 2c. A weakly stable layer is observed at 01:30 LST for both ShDeep and ShCu regimes. The potential temperature near the surface increases, from the nighttime to morning by about 1.0 K on ShDeep days while on ShCu days these changes are about 0.3 K only. At the morning stage (07:30 LST), a shallow mixed layer capped by a stable BL inversion above is observed due to the evolution of the convective layer and surface heating. At 10:30 LST, there are well-mixed layer conditions during the rapid growth stage for both regimes. At 13:30 LST, there is not a well-defined mixed layer on ShDeep convection days due to the effects of precipitation while a deepened well-mixed layer is found on ShCu days. From the stable, the morning transition, to the rapid growth stages on ShDeep days, the potential temperature profile is slightly cooler (about 1.0 K) than on ShCu days in the layer under 1000 m. In the cloudy mixing layer stage (13:30 LST), the colder low atmosphere during ShDeep days is also associated with precipitation caused by downdrafts, which may occur during the ascent period of the sounding on the subcloud layer.

The wind speed increases with the height, from the surface to 3000 m, except in the 07:30 LST sounding (Fig. 2d). The wind speed profiles show an overlap of large standard deviation in both convective regimes. These large standard deviations indicate high variability in the vertical wind speed profiles. The difference in horizontal winds between the two convective regimes is found during the morning transition stage (07:30 LST sounding). On ShDeep days, there is a maximum value of 12 m s$^{-1}$ at a height $\sim$ 1200 m. On ShCu days, the maximum wind value is approximately 10 m s$^{-1}$ at an altitude of 500 m. The contrast between the vertical variation of the wind suggests a larger wind shear on ShDeep days. Some of these features were also found by Zhuang et al. (2017), in which the 0-3000 m bulk vertical wind shear reaches a maximum of about 9 m s$^{-1}$ during the morning transition in the dry season considering the two years of the GoAmazon campaign (see Fig. 14 in Zhuang et al. (2017)). As a consequence, these larger values of vertical wind shear on ShDeep days might provide an elevated source of turbulent mixing (Mahrt and Vickers, 2002) during the morning transition that influences the growth of the convective BL.

The BL evolution is influenced by cooling and moistening in the first 1 000 m leading to a higher RH profile throughout the lower atmosphere. Temporal variations in temperature and moisture observed in the BL and lower free troposphere also resulted in considerable differences in convective indices. Figure 3 shows the evolution of the mean and standard deviation of the LCL, CIN and CAPE. Both LCL and CIN show a diurnal cycle due to the strong increase in diurnal cycle of BL temperature and moisture. The environmental condition is relatively moister than the environmental conditions on ShCu days leads to lower LCL and CIN on ShDeep days. ShCu days have lower free-tropospheric and BL moisture and thus higher LCL and CIN value (Zhang and Klein, 2010, 2013) creating a larger barrier for BL processes (Tian et al., 2021). On ShDeep days, the lower LCL associated with the rapid formation of convective BL results in cumulus formation (e.g., Zhang and Klein, 2013) earlier in the day. As expected for the end of the dry season, CAPE values are high for both ShDeep and ShCu days, and in particular, its maximum is observed around 10:30 LST the nearest time before the triggering of deep convection (e.g., Zhuang et al., 2017). The increase in humidity at low and mid-levels contributes to the change in atmospheric stability and thus to the triggering of the transition from shallow to deep convection. This result agrees with the findings of several related studies on the evolution of deep moist convection in the dry season during GOAmazon 2014/5 (Ghate and Kollias, 2016; Zhuang et al., 2017; Chakraborty et al., 2018).

## 3.2 Surface effects

### 3.2.1 Net radiative, soil moisture, and turbulent heat fluxes at the surface

Figure 4 illustrates how the BL controls cloud-surface interactions. The net shortwave radiative flux during the night is zero and, after sunrise, it increases to a maximum of $\sim 800$ W m$^{-2}$ at noon, and decreases in the afternoon (Fig. 4a). This is mainly modulated by the diurnal evolution of convective clouds. During the BL rapid growth stage, the main differences between ShDeep and ShCu days occur in the late morning due to the higher amount of shallow cumulus cloud cover on ShDeep days. On days with ShDeep, during the cloudy mixing layer stage, clouds are deeper, with larger optical depth, reflecting more shortwave radiation. Consequently, they reduce the shortwave radiation and cool the surface. This differences is very clear when ShDeep days are compared to ShCu days during the late morning or afternoon. ShDeep days have lower net shortwave radiation than ShCu days by an average value of around 70 W m$^{-2}$ between 09:00 and 10:00 LST, and 200 W m$^{-2}$ between 12:30 and 15:00 LST. The net longwave flux at the surface is always negative (Fig. 4b). The more negative the values are on ShCu days, the higher the outgoing longwave radiation is compared to ShDeep days. The difference between ShDeep and ShCu days is around $-10$ W m$^{-2}$ during the stable stage and increases during the day as the cloud fraction difference between them increases. The higher humidity profile in ShDeep days is probably the reason for this difference.

Soil moisture is an important factor because it controls the partitioning of the available energy between the surface latent and sensible heat fluxes. The mean soil moisture is lower on ShDeep days than on ShCu days in the superficial layer at 2.5 cm (Fig. 4c). In particular, there is a large overlap in standard deviation suggesting that there are days when the percentage of soil moisture is similar between the two convective regimes due to day-to-day variations. Since we found differences in the longwave radiative flux and soil moisture between ShDeep and ShCu days, differences in soil heat flux between convective

regimes are expected. This is shown in Fig. 4d, which represents the time evolution of the average soil surface heat flux for the soil layer $0-5$ cm. For the stable stage, the soil heat flux is positive (directed into the soil). On ShDeep days, the soil absorbs less energy than on ShCu days during nighttime. The morning transition stage is characterized by a crossover of the signal (positive to negative) around 8:00 LST in the morning in both convective regimes. The soil heat flux reaches a maximum of about $-35$ Wm$^{-2}$ around noon indicating an amount of released energy directed into the atmosphere. In the cloudy mixing

layer stage, the corresponding negative values and the large standard deviation on ShDeep days are associated with cooling by rain and the soil heat flux changes its flow direction around 16:00 LST. On ShCu days, the evening crossover occurs after sunset.

     The time evolution of the surface fluxes is in phase with the net shortwave radiation (Fig. 4a). The surface latent heat flux reaches a maximum of 290 W m$^{-2}$ on ShDeep days and 310 W m$^{-2}$ on ShCu days Fig. 4e. On ShDeep days, the latent heat

flux is lower than on ShCu days from 09:00 LST to 17:00 LST. This difference could be due to the early deepening of the BL, redistributing water vapor in the BL depth. The reduction of latent heat flux on ShDeep days is similar to that found over to the SGP site shown by Zhang and Klein (2010). At nighttime, due to surface longwave radiative cooling, the surface sensible heat is near zero or eventually slightly negative for both ShDeep and ShCu days (Fig. 4d). Early in the morning, the difference between ShDeep and ShCu days becomes significant after 07:00 LST, when the surface warming leads to an upward exchange

of sensible heat and subsequent warming of the lowest part of BL due to heat flux convergence. During the morning transition stage, the sensible heat flux reaches mean values of 20 W m$^{-2}$ greater on ShDeep days than on ShCu days. In the rapid growth stage, around 10:00 and 11:00 LST, the average sensible heat flux on ShDeep days reaches a peak of 50 W m$^{-2}$ higher than on ShCu days. The peak corresponds to the time before the triggering of deep convection. As expected, a drastic decrease of the surface heat flux is observed during precipitation events, around 13:00 LST with a decrease of almost $\sim 30$ W m$^{-2}$ and

around 16-17:00 LST around $\sim 70$ W m$^{-2}$.

     A pronounced feature of the diurnal evolution of the sensible heat flux, often seen on ShDeep days, is the effect of sensible heat flux promoting the erosion of nocturnal BL. After the crossover, the sensible heat flux increases with time and with the amount of energy released from the surface, driving a higher growth rate of the BL in the morning transition stage than on ShCu days. At the cloudy mixing layer stage, for ShDeep days, clouds affect the surface sensible heat flux, either positively

by increasing the sensible heat flux due to processes associated with the arrival of the gust front or negatively by diminishing the sensible heat flux due to the transport of cool air from aloft into the BL (e.g. Oliveira et al., 2020). On the other hand, the more entrainment of dry air from the free tropospheric into the BL on ShCu days may be a response to BL turbulence and more sensible heat flux during the cloudy mixing layer stage.

### 3.2.2   Surface inhomogeneity

The convective BL growth is forced by turbulent surface heat fluxes, which are controlled by instability and surface heterogeneity effects. Figure 5 shows the time evolution of the average air temperature at 2 m, specific humidity at 2 m, horizontal wind speed, and turbulent kinetic energy (TKE) and their standard deviation. The air temperature gradually decreases by 1.1°C and 2.3°C from 20:00 to 06:00 LST on ShDeep and ShCu nights , respectively. During the stable stage, the near surface air

temperature is controlled by longwave radiative cooling (Fig. 4b) and turbulent mixing. In the predawn hours, ShDeep days have higher surface temperatures on average (by about 0.5 °C) between 04:00 and 06:00 LST, than ShCu days. Followed the morning part of the air temperature evolution, which includes the morning transition stage and the rapid growth stage, there is an intense gradual warming of 8.0°C on ShDeep days and 10.0°C on ShCu days until the diurnal cycle reaches the maximum air temperature around noon for ShDeep days and around 15 LST for ShCu days. Later, on ShDeep days, after the air temperature reaches the maximum of 33.0°C around 12:00 LST, a drop of ~3.0°C is observed during the cloud mixing stage due to the latent cooling from rain evaporation . On ShCu days, during the cloudy mixing layer stage, the air temperature decreases after the maximum of 34.5°C around 15:00 LST.

Figure 5b shows the time evolution of the surface specific humidity, with similar behavior of air temperature. The specific humidity gradually decreases during the stable stage of the BL. For ShDeep days, the range is smaller than for ShCu days with rate values of 1.1 g kg$^{-1}$ during ShDeep nights compared to 2.0 g kg$^{-1}$ in ShCu nights in 8 h. By ~06:30 LST, the surface specific humidity is minimum followed by an increase of almost 3.0 g kg$^{-1}$ of moisture at 08:00 LST. After 09:00 LST, the specific humidity on ShDeep days decreases slightly in the afternoon by about 1.0 g kg$^{-1}$ between 12:00–16:00 LST at the time of precipitation. This drying of the BL on ShDeep days suggests that downdrafts are not able to bring air down from a high enough altitude to produce significant surface drying as it occurs in some cases of isolated convection (Oliveira et al., 2020) and also organized systems with system passages (Schiro and Neelin, 2018). While on ShCu days, the surface specific humidity is maximum around 08:00 LST followed by a drying throughout the rapid growth stage and cloudy mixing layer stage of around 2.0 g kg$^{-1}$ until the sunset, followed by a later maximum at 19:00 LST. These findings are consistent with those of Zhang and Klein (2013), where a large sensible heat flux during the cloudy mixing layer were observed to contribute to greater entrainment of dry air into the BL in ShCu in the SGP site.

In order to characterize the dynamical aspects of BL inhomogeneity, Fig. 5(c-d) shows the horizontal wind speed and turbulence intensity, which was checked by means of TKE. TKE is calculated as half of the sum of the variances of the wind components (TKE $= 0.5(\overline{u'^2} + \overline{v'^2} + \overline{w'^2})$). On ShDeep days, the mean nighttime surface wind speed is slightly larger than on ShCu days. For instance, from 22:00 to 02:00 LST, the period shows signs of intermittent turbulence, with winds becoming stronger (around of 1.5 m s$^{-1}$) and TKE peaks of ~0.3 m$^2$ s$^{-2}$, as expected since stronger winds imply increased mechanical production of turbulence at nighttime. Since the beginning of the convection, the daytime surface wind and the TKE increase. The TKE difference can be distinguish between the convective regimes during and after the passage of convective storms. Substantial enhancements of TKE are observed of about 1.5 m$^2$ s$^{-2}$ at 13:00 LST and about 0.5 m$^2$ s$^{-2}$ at 15:00 and 16:00 LST. These transient peaks of TKE are probably associated with convective storm downdrafts at the time the gust fronts arrived at the observational site (e.g. Oliveira et al., 2020).

### 3.3 Boundary layer effects

 #### 3.3.1 Comparing the boundary layer evolution between ShDeep and ShCu days

The daily evolution of the BL height is analyzed in Fig. 6 for ShDeep and ShCu days. It shows typical characteristics of the known Amazonian BL (Carneiro and Fisch, 2020), with minimum heights during nighttime and maximum around noon (Fig. 6, lines). During the night, the BL is established as an intermittent weakly stable layer, in which the BL depth decreases gradually from 20:00 LST to sunrise, ranging from 400 to 140 m on ShDeep days, and from 300 to 110 m on ShCu days. The average depth of the stable layer is higher on ShDeep days than on ShCu days (on average and standard deviation of $80 \pm 50$ m over the whole stable stage). The relatively higher values of nocturnal BL height are related to more turbulent mechanical mixing on ShDeep days than on ShCu days, as shown in Fig. 5d.

The morning transition stage begins at sunrise and is determined by the time when the surface heat changes from negative to positive values, that is $\sim$07:00 LST (Fig. 4d). When the surface warms after 07:00 LST on ShDeep days, there is a slight increase in the BL depth. This stage ends around 08:00 LST, at which time the net radiation is $\sim$240 W m$^{-2}$ and the sensible heat flux is about 39 W m$^{-2}$. Moreover, on ShCu days, the BL depth grows slowly from the approximately same time and ends 0.75 h later (around 08:45 LST), at which time the net radiation is $\sim$290 W m$^{-2}$ and the sensible heat flux is about 42 W m$^{-2}$. During this transition stage, the stable BL inversion is completed eroded and replaced with a convective BL. A noticeable difference between the convective regimes is that the period of the morning transition is shorter for ShDeep days than for ShCu days.

Throughout the rapid growth stage, the BL height advances rapidly through the morning. On ShDeep days, the beginning is observed 2 h after sunrise (end of morning transition stage) with a BL height of about $\sim$300 m and to continue to a height of $\sim$2000 m reaching through the residual layer around 09:30–10:00 LST (Fig. 6, dashed lines). During ShCu days, the BL takes 3 h after sunrise to extend from $\sim$300 m through the residual layer (around 2000 m). By the time the residual layer from the previous day is incorporated into the rapid growth, there is an increase in the buoyancy entrainment flux probably due to the moist lower troposphere (between surface and 1500 m ) (Chakraborty et al., 2018) which leads to a well-mixed layer supporting the formation of the deeper cloudy BL, on both convective regimes. ShDeep days reach a maximum height (in the mean) of $\sim$ 1800 m around 10:00 LST and then decrease in height during the emergence of shallow formation and during its growth into a deep cloud. ShCu days reach a maximum height (on average) of $\sim$ 2250 m around 13:00 LST and then decay in height in the late afternoon due to decreasing surface fluxes (Fig. 4e-f).

#### 3.3.2 The morning transition boundary layer and its relation to the afternoon ShDeep convection

Many processes can influence the BL throughout the diurnal cycle, such as surface heating, entrainment at the BL top, direct radiative heating or cooling of the air, and cloud effects (Angevine et al., 2020). Here, the dependence of meteorological parameters and processes on the BL associated with a more rapid transition to the convective onset is shown for each ShDeep and ShCu event. As noted previously, the response of the convective regimes depend on the tropospheric state, more specifically on the integrated-column humidity (Ghate and Kollias, 2016), and on the BL processes (e.g. Zhang and Klein, 2010). On

ShDeep days, the nocturnal BL evolution during the stable stage is characterized by more mixing and less time to erode the stable BL inversion, leading to an early well-mixed layer that favors the rapid formation of the convective BL compared to ShCu days. During the morning transition stage, humidity and VWS can be relatively large in the 1000 m layer as well as the
360 sensible heat flux at the surface. This suggests that some of the warming that is eroding the stable BL may be associated with surface and top-down effects, similar to previous findings on the morning transition (e.g., Angevine et al., 2001, 2020).

The main characteristics of the morning transition of all days are given in Table 2 (a variation of this table can also be found in Fig. 7). The integrated water vapor (IWV) is calculated in the column from 50 to 1000 m (IWV$_{1km}$), and in the total vertical column, from 50 to 20000 m (IWV$_T$), and are shown in brackets (Table 2). The vertical wind shear (VWS) is calculated by
365 subtracting the mean horizontal wind speed at 1000 m from the mean 50 m wind speed at 7:30 LST radiosonde profile. Days with cloud influence during the morning-transition stage are marked with an asterisk in the table. Angevine et al. (2001) has shown that the most extreme day-to-day variations in the duration of the morning transition were found on days with cloud influence at Cabauw tower in the Netherlands. The daytime convective BL over Amazonia is generally cloud influenced and rarely cloud-free (Betts et al., 2009), therefore days with cloud influence during the morning transition stage are not excluded
from the analysis.

On average, the duration of the morning transition stage for the four ShCu days is about 2.5–3.0 h after sunrise and the specific humidity decreases sharply from $\sim 19$ to 12 g kg$^{-1}$ for the layer between 50 and 1000 m as reported by the soundings profile. Among ShCu days, day 1 (25 August) and day 4 (4 September) are cloud-free days. The time required for the sensible heat flux to cross over zero is 1.5 h and the onset of the convective BL is observed about 2.5 h after sunrise for both days. These
375 two days have respectively the greatest and the smallest height rate-of-change among ShCu days. The height rate-of-change is 84 m h$^{-1}$ for day 1 and 43 m h$^{-1}$ for day 5. The difference between these days is probably due to the larger sensible heat flux at the time of the crossover and the vertical structure of the specific humidity as well as the IWV$_T$ of 4.6 cm for day 1 compared to 3.6 cm for day 4. The days has moderate VWS of 5.1–4.8 m s$^{-1}$. For both ShCu days with cloud influence, day 3 (3 September) has the earliest crossover among ShCu days of around 0.5 h after sunrise, and has the last CBL onset
at 09:00 LST, 3.0 h after sunrise probably due the shallow cumulus cover observed between 07:00 and 08:00 LST. Day 5 (5 September) has a low height rate-of-change (56 m s$^{-1}$) and the smallest VWS of all days (3.2 m s$^{-1}$). The crossover occurs at 1 h after sunrise and the onset is 2.5 h after sunrise.

Based on the ShDeep days, low cloud cover is present on 3 of the 4 days. The average duration of the transition is about 2.0 h for the cloud-free day and days with shallow cloud influence. On ShDeep day with shallow and altocumulus cloud influence
during the whole morning transition stage, the duration is about 2.5 h. In all ShDeep days, the specific humidity decreases from $\sim$20 to $\sim 15$ g kg$^{-1}$ for the layer between 50 and 1000 m. On day 2 (27 August), shallow cumulus and altocumulus are reported before and after sunrise, however, the sunrise follows the crossover by 0.5 h and after that, between the crossover and the onset, there has no influence of clouds and the onset of convective BL is approximately 2.25 h after sunrise. For the day 3 (7 September), shallow clouds are observed between 06:00 and 07:30 LST, the crossover is 1.0 h (07:00 LST) after sunrise and
the onset follows at around 08:10 LST. Day 7 also has a high BL height rate-of-change like day 2 and day 6 of about 200 m s$^{-1}$. However, the onset of convective BL is delayed by 2.5 h after sunrise probably due to altocumulus and cumulus cover

during the transition. The last selected ShDeep day is 12 September, the only cloud-free morning and the smallest BL height rate-of-change and VWS among ShDeep days. The sensible heat flux crossover time is 06:30 LST and the onset occurs 2 h after sunrise (08:00 LST).

The early onset of the convective BL is visible on ShDeep days as a clear increase in turbulence (about $0.1 \mathrm{~m\,s^{-1}}$) in the standard deviation of the vertical wind speed contour (Fig. 8). Thus, there is a good agreement of the early onset observed between SODAR and the ceilometer on ShDeep days (Fig. 8e-h). Furthermore, the SODAR profile onset time is a little later on ShCu days than ShDeep days of approximately 0.5–0.1 h (Fig. 8a–c) as observed by the ceilometer.

After detecting the large VWS in the morning transition sounding, we found that the VWS above the BL is generally associated with the occurrence of low-level jet (LLJ) stream in 7 of the 8 days (Table 2). The exception is day 8 on 12 September, which shows a VWS of $7 \mathrm{~m\,s^{-1}}$. The LLJ is observed as a low-level maximum speed in the vertical wind profile, in which the wind is faster than the wind speeds above and below it in the lowest 1500 m of the atmosphere (Stull, 1988). On ShDeep days, the maximum wind speed is $12–15 \mathrm{~m\,s^{-1}}$ located between 630 and 1350 m above the ground. On ShCu days, the wind speed profile indicates the development of a LLJ with the nose (maximum wind speed) of the jet located at a height of around 350–630 m, with a maximum speed of about $6–13 \mathrm{~m\,s^{-1}}$. The deeper LLJ on ShDeep days may be associated with large-scale moisture advection above the BL as showed by Ghate and Kollias (2016).

The scatterplots in Fig. 9 show the relation between morning transition BL height rates-of-change and (a) $IWV_{1km}$ and (b) VWS between 50 m and 1000 m, respectively. The BL height increases with time due to the surface heating. It indicates a positive association of a high rate-of-change of the BL height due to the high values of $IWV_{1km}$ and VWS (e.g., the duration of the morning transition on days with ShDeep convection decrease at high humidity content and at intense VWS in the early morning, both for cloud-free and cloudy-influenced mornings).

The main findings are summarized in the conceptual schematic shown in Fig. 10 for the cloudy-BL stages when the occurrence of only shallow cumulus is favored, while in others, shallow cumulus clouds evolve into deep convective clouds.

## 4  Boundary layer processes statistics during the two dry seasons of the GoAmazon 2014/5 campaign

In order to evaluate the significance of the conceptual model built based in the case studies, a statistical analysis was performed. The statistical analysis was developed by comparing the cases of shallow and deep convective clouds using the same classification and relaxing the premise of having convection under the T3 site. ShDeep days were classified as hot-tower clouds with little external disturbances in the morning based on cloud boundaries of the RWP-WACR-ARSCL product and the precipitation properties for the domain within the 75 km radius of the T3 site estimated by S-band Radar scanning (Tian et al., 2021). ShCu days were classified as days with no precipitation in the 75 km radius of the T3 site. Using the two dry season years, 45 cases of ShDeep and 30 cases of ShCu were selected. On the majority of days, the shallow-to-deep cloud evolution did not occur at the T3 site but in the 75 km radius domain of S-Band Radar. The statistic analysis of the BL processes mainly focuses on the stable to the morning transition stages as it has been shown in the conceptual model of the BL process (Fig. 10). This large number of events is robust to describe the daily evolution of a typical day of ShCu and ShDeep.

The composites of the BL processes, discussed in detail in the previous section, are shown in Fig. 11 as box-and-whisker plots. As previously discussed, Ghate and Kollias (2016) suggested that greater fluctuation over short periods in IWV is the primary factor controlling precipitation during the dry season. For the dry seasons of two years, higher IWV and VWS were found in the layer between surface and 2 km than 1 km as in the case studies, reflecting the variability through the dry season progression. In fact, the median $IWV_{2\ km}$ composite, at the lower 2 km, shows larger values and exhibits smaller variability

in the ShDeep (blue boxes) than ShCu (red boxes) regimes since nighttime (stable stage) (Fig. 11a). In the lower troposphere, distinct variability in ShDeep conditions is also observed in $VWS_{2\ km}$ from the stable stage to the morning transition stage, with larger median values than that for ShCu (Fig. 11b). Zhuang et al. (2017) found a similar result, concluding that the contrast between ShDeep and ShCu days could be associated with weak surface wind and strong wind shear in the lower troposphere. Moreover, this result indicates that turbulent mixing may be transported from aloft down to the surface, linking to the evolution

from stable to unstable BL conditions.

During the stable stage at near surface, less radiative cooling and intense periods of turbulent mixing characterized the ShDeep composite in comparison with ShCu. The net longwave showed very similar results to the case studies (Fig. 11c) with a well-defined difference of around 5 W m$^{-2}$ between the ShDeep and ShCu composites. The median shows more intense radiative cooling during the ShCu regime than for ShDeep. For example, Edwards et al. (2014) using large-eddy simulations

showed that radiative cooling has an important effect on the morning transition and in the development of the convective BL. The mechanical turbulence is seen in Fig. 11d by the TKE. The magnitude of the TKE is comparatively higher for the ShDeep between 00:00 and 03:00 LST than those of the ShCu composite. Similar small median values occur at around 06:00 LST for both regimes when the increase in turbulence is weak, probably caused by weak wind at late night and cumulative thermal cooling inducing stable stratification near the ground through the nighttime.

In contrast to the longwave radiative cooling, the sensible heat flux showed a less pronounced difference with increasing sample size, being slightly higher in the median ShDeep values than in the median ShCu values during the crossover event (Fig. 11e). One explanation for this difference is that the sensible heat flux is affected by cloudiness and the surface thermal properties, such as the one shown in Fig 1g, on day 7 of the case studies. In that case, the sensible heat flux is influenced, before or during the morning transition, by reducing incident solar radiation, surface temperature, and therefore its magnitude and time

crossover (e.g reduce the intensity of convection). However, considering that under ShDeep conditions there is less radiative cooling, a small variation in the sensible heat flux at the time of crossover can resulting in early erosion of the nocturnal inversion that develops during the stable stage. Such cloud cover is less frequent in the ShCu regime during the early morning, if clouds are more frequent in the afternoon period attributed to the shallow cumulus cloud type (Giangrande et al., 2020).

During the stable stage, the ShDeep BL height is higher than that of the composite ShCu. The relatively high BL height

values are consistent with relatively higher mechanical turbulence as the source of TKE (Fig. 11d). The hourly median BL height values from sunrise to early morning vary from about 110 to 250 m between 07:00 and 08:00 LST in the ShDeep regime (Fig 11f). For the ShCu regime, the median hourly BL height values vary from around 110 to 200 m, suggesting that the onset of convective BL is earlier in the ShDeep composite regime than in the ShCu composite regime. The BL height evolution exhibited in both regime composites is consistent with the evolution observed in the case studies during the morning transition.

The statistical analysis is in agreement with the conceptual model developed based on the case studies, even do not considering convection over the T3 site. Also, the statistical two-sided Student's *t* test showed that statistically significant differences are present for all times of IWV, VWS, longwave radiative cooling, and TKE only for at 3 LST at the 95% confidence level between ShDeep and ShCu days. The BL height during the morning transition was statistically significant at the 95% confidence level at 7LST, at the 89% confidence level for 8 LST consistent with the difference of erosion time of BL between ShDeep

and ShCu days. The pronounced feature that emerges from this analysis is the clear difference between the two regimes for IWV and VWS and the BL height. This impact in the time needed to erode the nocturnal stable BL inversion used to promote convection having an important effect in characterizing the convective strength in the morning transition stage and the rapid growth stage. This energy of BL growth is in agreement with the large, positive BL disequilibrium values on ShDeep days and the mostly negative BL disequilibrium values in early morning on ShCu days showed by Tian et al. (2021).

During the dry season, the trigger mechanism of ShDeep is a local effect. However, it only happens when there is enough IWV and non-local factors control this. The IWV for ShDeep days is explained by the synoptic patterns as suggested by Biscaro et al. (2021). Also, Ghate and Kollias (2016) shows that large-scale moisture advection can control the relationship between local land-atmosphere interactions and diurnal precipitation. Here, we show that the enhanced water vapor at low and mid-level is important for the characterization of the cloud-BL interaction from the stable to the rapid growth stage and

triggering ShDeep in the cloudy mixing layer stage. This feature, observed during the dry season over Central Amazonia, was corroborated by Chen et al. (2020) who showed that large-scale advection is vital for characterizing land-atmosphere interactions both in magnitude and type of relationships in the transition from clear-sky to precipitation clouds over the U.S. SGP site.

## 5   Conclusions

This study presents an observational analysis of Amazonian atmospheric BL processes from the nighttime stable stage until the cloudy developed stage during the dry season. The data used here were collected during the Green Ocean Amazon 2014/5 (GoAmazon) field campaign. These data are used to study the BL evolution for ShCu and ShDeep days for case studies and statistical analysis. The detailed analysis of the case studies allowed us to develop a conceptual model to describe the BL evolution from clear to cloudy BL and set up the most important parameters characteristic of each stage and the differences

among the convective day types.

The main findings are summarized in the conceptual model shown in Fig. 10. The environmental conditions (e.g., in the lower troposphere) in which the BL grows during ShDeep days are relatively moister and colder than in the environmental conditions in which ShCu days are observed. Analysis of the vertical structure of wind speed shows that VWS is generally associated with the occurrence of maximum wind speed (e.g., low-level jet). During the stable stage, the combination of

environmental conditions and nocturnal BL characteristics favor such marked variations observed in the morning transition stage. The duration of the morning transition stage on ShDeep days decreases when humidity is high, and VWS is intense in

the early morning hours. This rapid transition to convective onset can be attributed to a combination of variables before sunrise, such as surface specific humidity, warm air temperature controlled by radiative cooling, and turbulent mixing.

The early erosion and early peak of the rapid growth stage respond to changes in the partitioning between surface heat fluxes by increasing the surface sensible heat flux, which will force the growth of BL. This suggests that the morning transition of the nocturnal BL does play a role in the vertical motion in the convective strength on ShDeep days. In consequence, the onset of full convection (e.g., well-mixed layer) at the end of the morning transition stage occurs earlier than for ShCu days. During the rapid growth stage, subcloud lifting processes in the deeper BL will entrain moist air into the BL through which the air parcel can reach the LCL (thus causing the first cell of shallow cumulus to emerge) and then the level of free convection (thus favoring deep convection).

Our study shows that the enhanced water vapor at low and mid-level is at the origin of the main modifications of the BL processes since nighttime and early morning stages. In addition, our results show that the time taken to eliminate the nocturnal stable boundary layer inversion is used to promote convection having an important effect in characterizing the convective strength in the morning transition stage. The BL processes statistics show that the morning boundary layer conditions in the case studies are representative during the two dry seasons of GoAmazon, despite the case-to-case variability. This finding supports the previous results (Ghate and Kollias, 2016; Zhuang et al., 2017; Tian et al., 2021), but extends the knowledge by showing in detail the different BL processes evolved in the shallow to deep convection transition. In terms of numerical modeling for a realistic simulation of convective initiation, in both weather and climate modeling, is important a realistic representation of BL meteorology. Besides, the continuous increase in model resolution remains essential for an accurate and detailed description of the different physical processes during the daily transitions in the convective BL and for the improvement of atmospheric BL parametrization (e.g. Sorbjan, 2007; Edwards et al., 2020). A detailed analysis using large-eddy simulations to describe these BL processes, entrainment fluxes across the top of the convective BL, and turbulent kinetic energy budget on shallow and shallow-to-deep events will be the subject of a further study.

*Data availability.* The data used in this study are available in the ARM Climate Research Facility database for the GoAmazon 2014/5 experiment (https://www.arm.gov/research/campaigns/amf2014goamazon)

*Author contributions.* AH performed the analyses, and all the authors contribute to the writing of the manuscript.

*Competing interests.* The authors declare that they have no conflict of interest.

*Acknowledgements.* We acknowledge the data from the Atmospheric Radiation Measurement (ARM) Program sponsored by the U.S. Department of Energy, Office of Science, Office of Biological and Environmental Research, Climate and Environmental Sciences Division. AH acknowledges the Brazilian National Council for Scientific and Technological Development (CNPq) graduate fellowship (grant agreement number 140347/2019-4) and the EU FET VESTEC H2020 project, grant agreement number 800904. The authors GF (process 307048/2018-7) and LATM (process 305301/2017-9) also thanks to CNPq for their Bolsa de Produtividade em Pesquisa. This study was financed in part by the Coordenação de Aperfeiçoamento de Pessoal de Nível Superior - Brasil (CAPES) - Finance Code 001. We also acknowledge FAPESP CHUVA Project Grant 2009/15235-8. We also thank Thiago Biscaro for his help in providing data from the S band Radar, Rayonil Carneiro for the discussion about ceilometer data, Daiane Brondani for her help with ECOR data, and Yang Tian for share the deep convection dates for our ShDeep days' comparison. Further, the authors would like to acknowledge two anonymous reviewers for their valuable comments.

*Financial support.* This work has received funding from the Brazilian National Council for Scientific and Technological Development (CNPq grant agreement number 140347/2019-4) and from the EU FET VESTEC H2020 project, grant agreement number 800904.

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

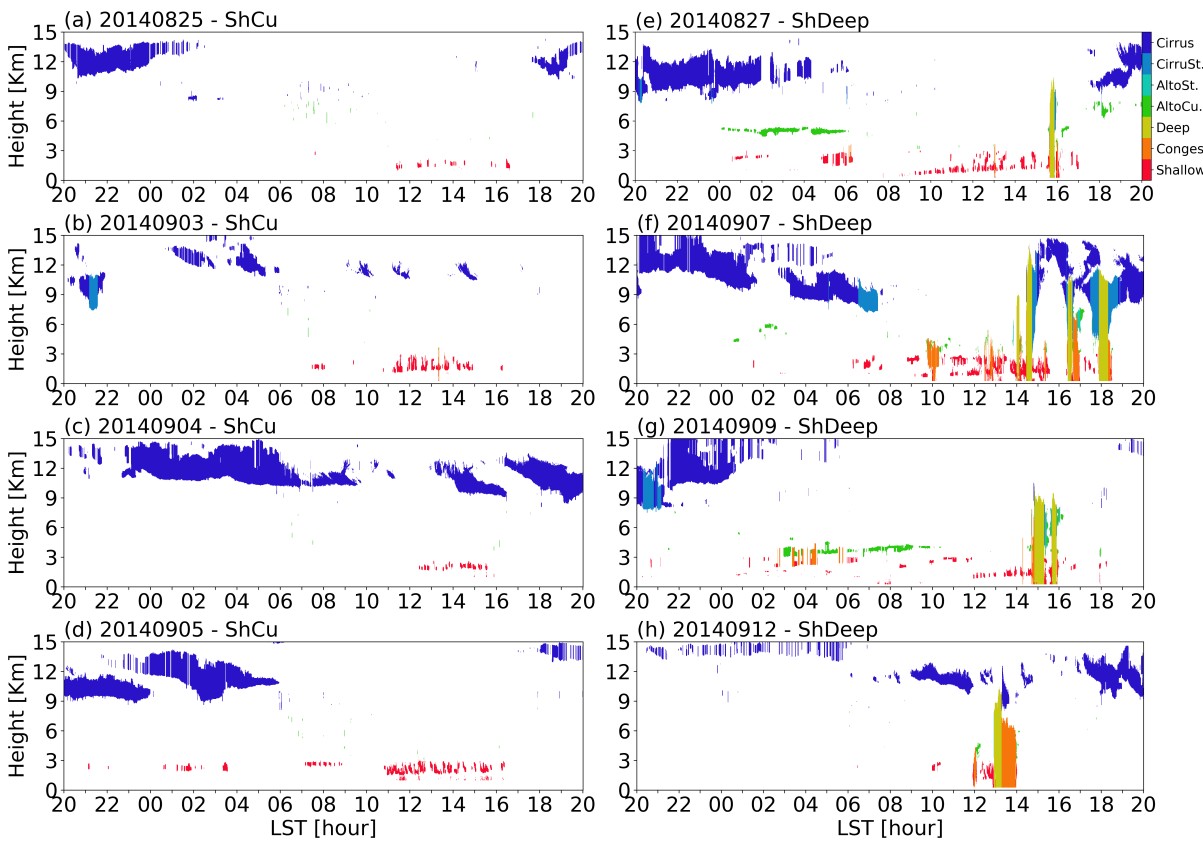

**Figure 1.** Cloud-type classification from the ARSCL product for corresponding dates of **(a–d)** shallow convective (ShCu) days on the left-hand panels and **(e–h)** shallow-to-deep convective (ShDeep) days on right-hand panels.

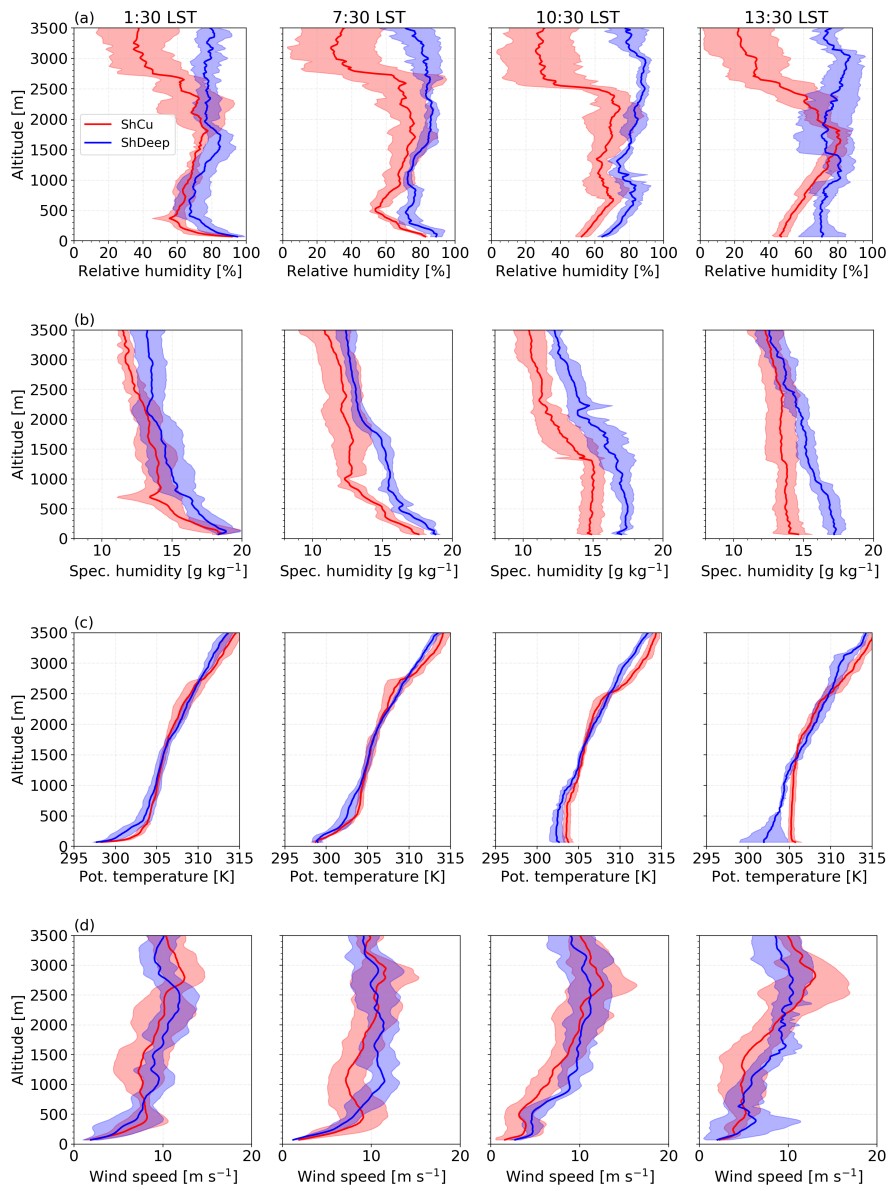

**Figure 2.** Vertical profiles between 50 and 3500 m of **(a)** relative humidity, **(b)** specific humidity, **(c)** potential temperature, and **(d)** wind speed from radiosondes launched at the T3 site (from left to right) at 1:30, 7:30, 10:30, and 13:30 LST. The mean (bold lines) and the standard deviation (shadings) are shown for shallow convective (ShCu, red) and shallow-to-deep convective (ShDeep, blue) days.

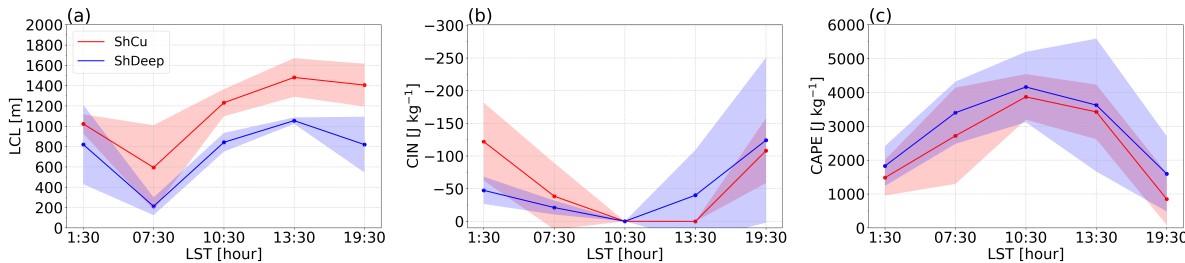

**Figure 3.** Time evolution of **(a)** lifting condensation level (LCL), **(b)** convective inhibition (CIN) and **(c)** convective available potential energy (CAPE) derived from radiosondes. The mean (bold lines) and the standard deviation (shadings) are shown for shallow convective (ShCu, red) and shallow-to-deep convective (ShDeep, blue) days.

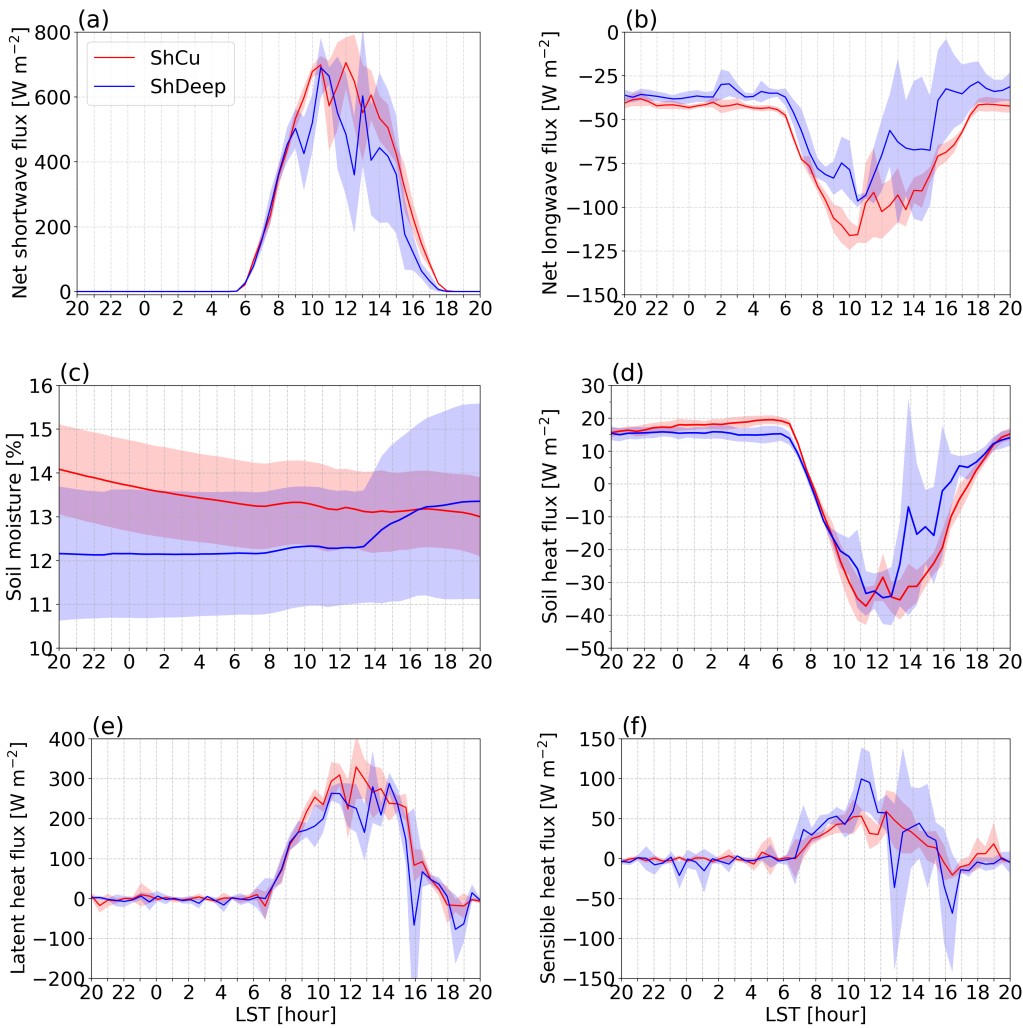

**Figure 4.** Time evolution of **(a)** net shortwave radiative flux, **(b)** net longwave radiative flux, **(c)** soil moisture at superficial layer, **(d)** soil heat flux, **(e)** latent heat flux and **(f)** sensible heat flux at the surface. The mean (bold lines) and the standard deviation (shadings) are shown for shallow convective (ShCu, red) and shallow-to-deep convective (ShDeep, blue) days.

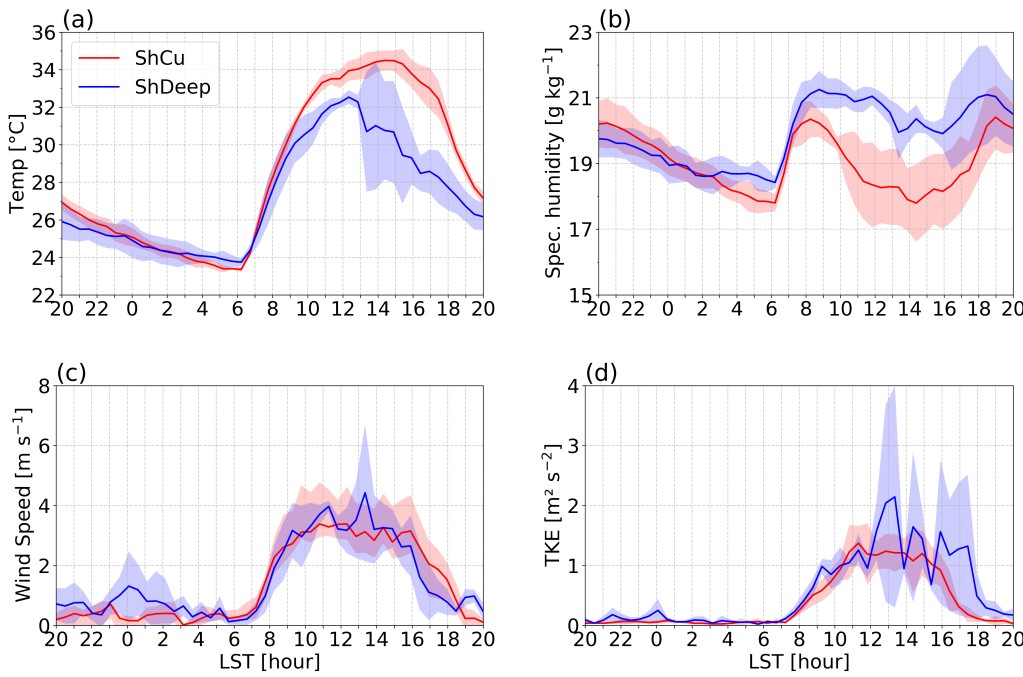

**Figure 5.** As in Fig. 4 but for **(a)** surface air temperature, **(b)** specific humidity, **(c)** wind speed estimated by surface meteorological system, and **(d)** turbulent kinetic energy estimated by ECOR.

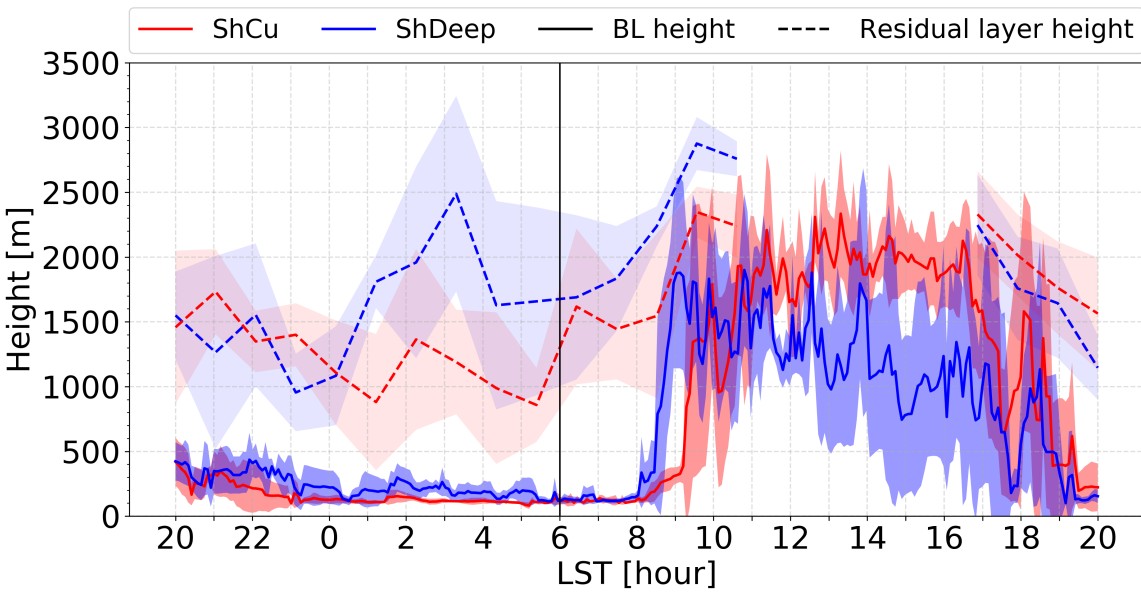

**Figure 6.** As in Fig. 4 but for the height of the BL (solid) and the residual boundary layer height (dashed).

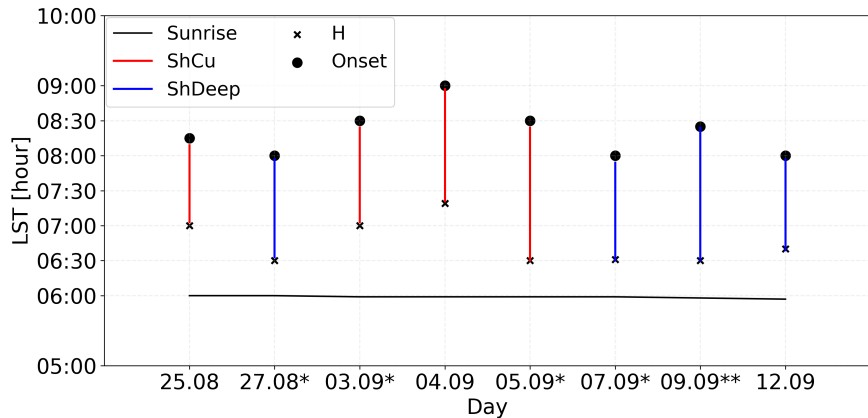

**Figure 7.** Times of sunrise (line), sensible heat flux crossover (H, "x"), and onset of convective BL estimated by ceilometer (Onset, circles) corresponding to eight morning transition days. The red and blue lines correspond to shallow convective (ShCu) and shallow-to-deep convective (ShDeep) days, respectively. The days with shallow cloud influence is marked as asterisk and day AltoCumulus cloud influence is marked as double asterisk.

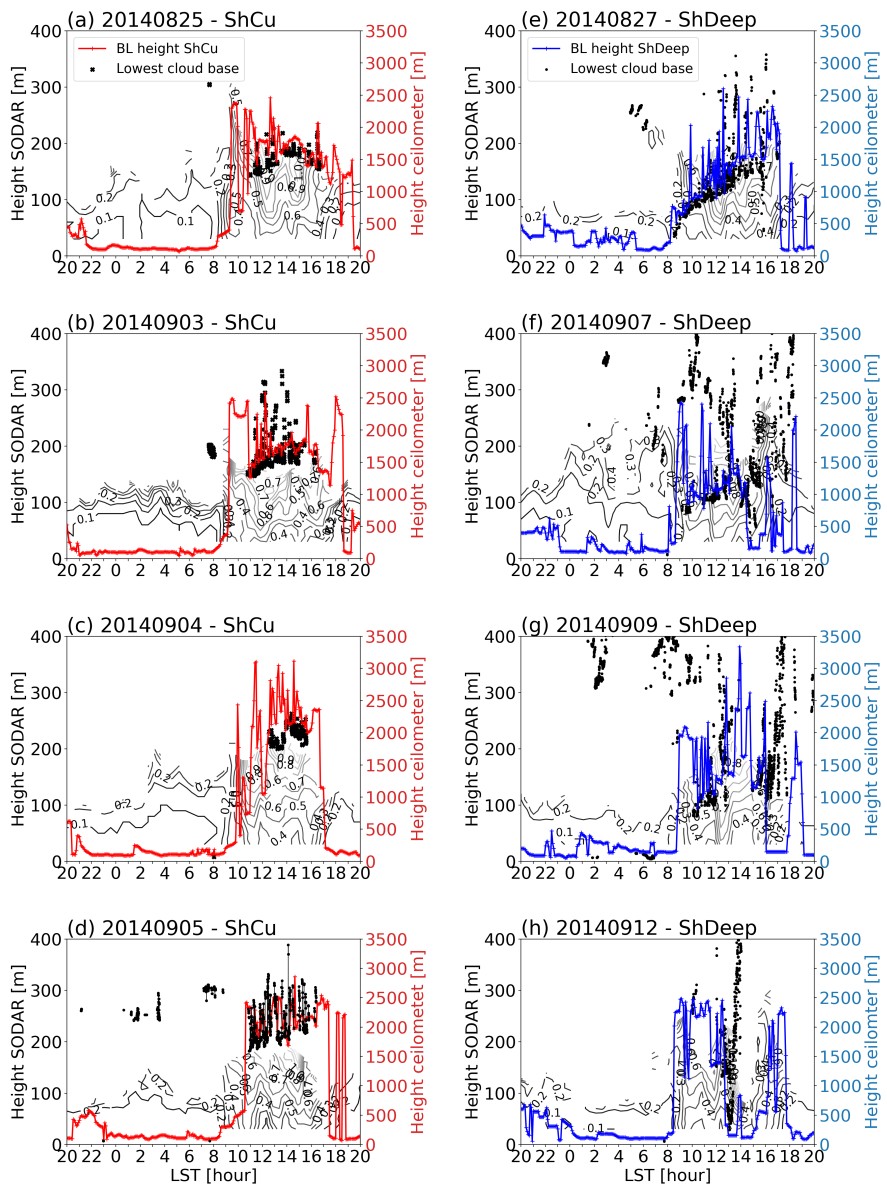

**Figure 8.** As in Fig. 4 but for the height of the BL and lowest cloud base (black dots) from ceilometer on **(a–d)** ShCu days and **(e–h)**ShDeep days. The contour shows the standard deviation of vertical velocity estimated by SODAR. The contour starts at 0.1 and increments every 0.1 m s$^{-1}$.

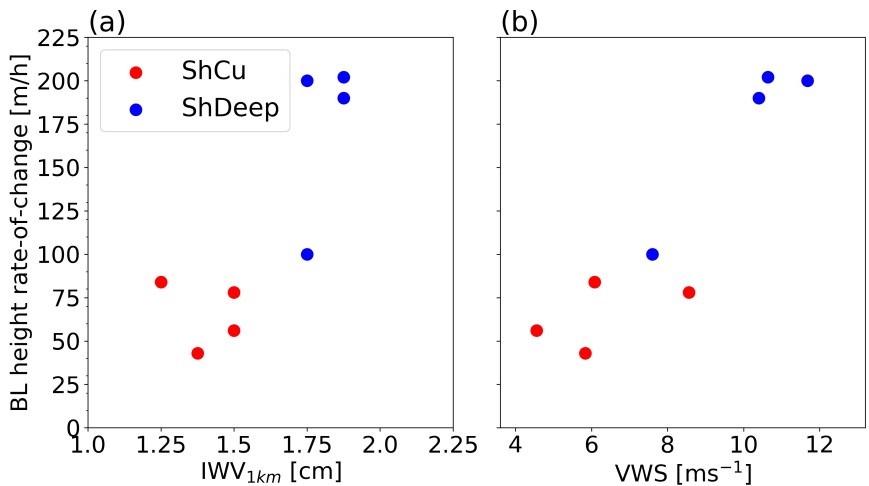

**Figure 9.** Scatterplots of the BL height rate-of-change versus **(a)** integrated water vapor from 50 to 1000 m ($IWV_{1km}$) and **(b)** vertical wind shear (VWS) between 50 m and 1000 m for shallow-to-deep convective (ShDeep, blue circles) and shallow convective (ShCu, red circles) days.

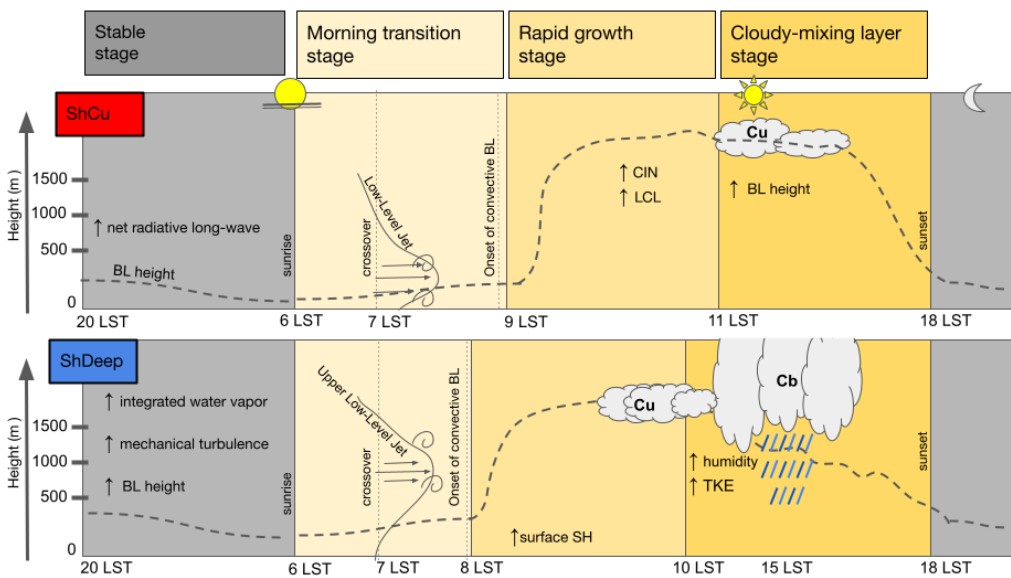

**Figure 10.** Conceptual model of BL processes on ShCu (top) and ShDeep days (bottom) illustrating the cloudy-BL stages during the dry season over Central Amazon. The BL height is represented by the dashed curves on both regimes. BL, SH, CIN, LCL, and TKE stand for the boundary layer, sensible heat flux, convective inhibition, lifting condensation level, and turbulent kinetic energy respectively.

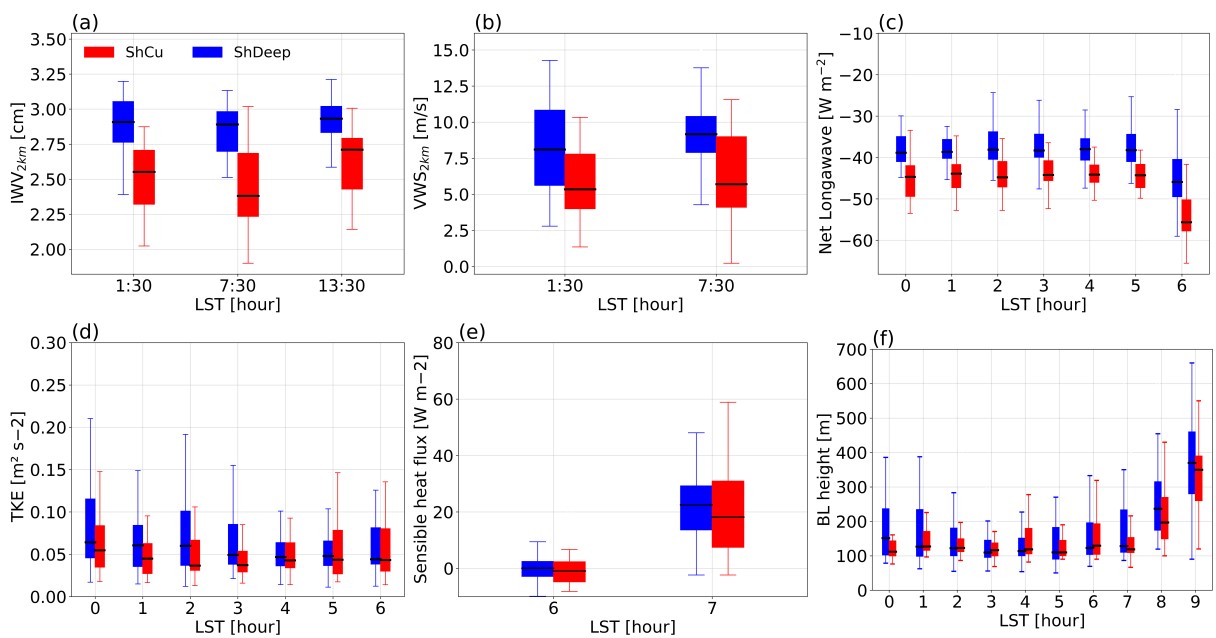

**Figure 11.** Box-and-whiskers plot of (a) IWV$_{2\ km}$ during the stable, morning transition, and cloudy mixing layer stages, (b) VWS$_{2\ km}$ at stable stage and morning transition, (c) radiative cooling during the stable stage, and (d) TKE during the stable stage, (e) sensible heat flux crossover, and (f) BL height from stable to onset of convective BL. In each box, the central bar is the median and the lower and upper limits are the first and the third quartiles, respectively. The lines extending vertically from the box indicate the spread of the distribution with the length being 1.5 times the difference between the first and the third quartiles. The red and blue boxes correspond to shallow convective (ShCu) and shallow-to-deep convective (ShDeep) days, respectively.

**Table 1.** List of non-transition shallow convective (ShCu) and transitions shallow-to-deep convective (ShDeep) days selected during the IOP2.

| Day | Date | Convective regime | Event Description | Precipitation rate [mm h$^{-1}$] |
|---|---|---|---|---|
| 1 | 2014/08/25 | ShCu | Scattered shallow cumulus convective cells | 0 |
| 2 | 2014/08/27 | ShDeep | Scattered shallow-to-deep convective cells | 4.0 |
| 3 | 2014/09/03 | ShCu | Scattered shallow cumulus convective cells | 0 |
| 4 | 2014/09/04 | ShCu | Scattered shallow cumulus convective cells | 0 |
| 5 | 2014/09/05 | ShCu | Scattered shallow cumulus convective cells | 0 |
| 6 | 2014/09/07 | ShDeep | Scattered shallow-to-deep convective cells | 3.3 |
| 7 | 2014/09/09 | ShDeep | Scattered shallow-to-deep convective cells | 3.4 |
| 8 | 2014/09/12 | ShDeep | Scattered shallow-to-deep convective cells | 4.7 |

**Table 2.** Meteorological parameters for each case study day during the morning-transition stage. Reference time of first positive significant value of sensible heat flux (Crossover); Time to erode the nocturnal BL relative to sunrise (Onset CBL); Change in BL height during morning transition stage (rates-of-change); The 50 m value of specific humidity ($q_{surf}$); Integrated Water Vapor from 50 to 1000 m ($IWV_{1km}$); total column IWV of are reported in parentheses ($IWV_T$), Vertical Wind Shear between 1000 and 50 m ($VWS_{1km}$); Low-level jet maximum (LLJ) and in parentheses are also reported the respective maximum heights. The $q_{surf}$, IVW, VWS, LLJ are computed using the 07:30 LST radiosoundings. Days with cloud influence during the morning transition stage are marked with an asterisk.

| Event | Date | crossover time [h](value) | Onset CBL [h] | rates-of-change [m h$^{-1}$] | $q_{surf}$ [g kg$^{-1}$] | IWV [cm] | $VWS_{1km}$ [m s$^{-1}$] | LLJ [m s$^{-1}$](height) |
|---|---|---|---|---|---|---|---|---|
| 1 | 2014/08/25 | 07:30 (20.0 Wm$^{-2}$) | 2:30 | 84.0 | 19.6 | 1.2 (4.6) | 5.1 | 11.3 (330 m) |
| 2* | 2014/08/27 | 06:30 (2.0 Wm$^{-2}$) | 2:15 | 190.0 | 19.6 | 1.7 (4.9) | 10.5 | 15.0(1350 m) |
| 3* | 2014/09/03 | 06:30 (3.0 Wm$^{-2}$) | 3:00 | 78.0 | 18.7 | 1.5 (4.4) | 8.2 | 13.7 (630 m) |
| 4 | 2014/09/04 | 07:30 (17.0 Wm$^{-2}$) | 2:30 | 43.0 | 20.0 | 1.4 (3.4) | 4.8 | 8.6 (350 m) |
| 5* | 2014/09/05 | 07:00 (0.1 Wm$^{-2}$) | 2:30 | 56.0 | 20.4 | 1.3 (3.6) | 3.2 | 6.6 (610 m) |
| 6* | 2014/09/07 | 07:00 (23.0 Wm$^{-2}$) | 2:10 | 202.0 | 20.5 | 1.7 (5.1) | 10.8 | 12.7(890 m) |
| 7* | 2014/09/09 | 07:00 (44.0 Wm$^{-2}$) | 2:30 | 200.0 | 20.1 | 1.6 (5.1) | 12.1 | 13.2(1010 m) |
| 8 | 2014/09/12 | 06:30 (8.0 Wm$^{-2}$) | 2:00 | 100.0 | 21.0 | 1.6 (4.6) | 7.0 | non-LLJ |

non LLJ = no LLJ is detected