# Peer review of "Morning boundary layer conditions for shallow to deep convective cloud evolution during the dry season in the Central Amazon"

_Atmospheric Chemistry and Physics, 2021_

## Author Comment (AC1)

Responses to referee comments on the acp-2021-87 Manuscript "Morning boundary layer conditions for shallow to deep cloud evolution during the dry season in the central Amazon", by Henkes et al.

**Referee Comments 1**

This paper provides a comprehensive study contrasting meteorological variables between shallow and shallow-to-deep convection transition cases, with a focus on boundary layer processes, using the GoAmazon field campaign data collected during the dry season (IOP2) of 2014. Some results are overall consistent with previous studies. However, there are two major issues for this study, one is new findings are not clearly presented and discussed, and the other one is the very small sample size. And the overall writing and English should also be improved. My recommendation is a rejection but I encourage a re-submission after these issues are addressed.

We thank you for your constructive comments and suggestions to improve our study. We have carefully addresses all comments/suggestions made by reviewer. Please find below our point-by-point responses in red color, and the changes to the manuscript are in blue color response.

1. After reading through the manuscript, one major issue is that I'm confused about what's new findings in this study. Many mechanisms related to BL processes and development are either well taught in university textbook or discussed in previous studies. Mechanisms or results unique to this study should be clearly discussed in either results or discussion section.

   Thank you for your suggestion. As far as we know, the BL evolution, based on stages of the development of convective cloudy-BL, has been shown only for undisturbed conditions over Amazonia (e.g., Martin et al. 1988; Fisch et al., 2004; Carneiro and Fisch, 2020). A detailed analysis of the erosion of nocturnal BL and growth of the convective boundary layer has not yet been reported under shallow-to-deep conditions. This erosion is important because the timing of increased convective turbulence in the morning transition is essential for the convection to penetrate throughout the BL, the transports of aerosol, and the formation of cumulus clouds upper BL.

   The main findings are shown in the conceptual model under ShCu and ShDeep conditions, which were not discussed in detail in the previous work. As suggested by both referees, we included a new section with statistical analysis of BL processes, especially during stable and morning transition stages (also note our response to the specific comment below). In summary, we rewrite part of the introduction, and we include a new section with BL processes statistics (Sec. 4) and rewrite the conclusions to be more focused on the new findings.

2. A big limitation of this study is the very small sample. With only four samples for each category, it's very hard to convince the readers that the average profile and the standard deviation capture the whole picture. The large error bars in many figures render many differences between two categories insignificant and thus some statements related to comparisons are not backed up by the figures. The authors argue that their samples show certain variations and are consistent with previous studies. To me, however, a few samples being consistent with previous studies (Line 403-405) cannot be used to justify why they are not using a larger sample when they can. I understand the GoAmazon only has very limited data in the IOP period for dry season, however, as far as I know, the main difference between IOP2 and non-IOP periods is only the additional 10:30 sounding, which only affects the results of Figure 2 and part of Figure 3. Thus, I think data of two whole dry seasons during GoAmazon should be utilized. On the other hand, the criteria for selecting ShCu and ShDeep can be modified as well to incorporate more samples. For example, shallow convection that did not directly pass over the site cannot be observed by ARSCL, but it can be shown in S-band radar. Also, it's also not clearly explained why the authors only focus on the dry season.

   As suggested by both referees, we have included a new section with BL processes statistics shown in the conceptual model for 45 ShDeep and 30 ShCu cases. The BL process statistics are discussed during the two dry seasons of the GoAmazon campaign (Sect. 4). Despite the case-to-case variability, the statistic analysis shown that the analyzes done in the case studies are representative of the cases for the dry seasons in Central Amazonia ((Line 460-514):

[revised manuscript text omitted]

105 Also, the manuscript was amended in some parts of introduction in order to be consistent with the manuscript purpose (comparative analysis of the BL processes under ShCu and ShDeep conditions during the dry season). For example Line 68-79:

Although previous studies, such as those by Ghate and Kollias (2016); Tang et al. (2016); Zhuang et al. (2017); Chakraborty et al. (2018); Tian et al. (2021) provide observational evidences and explanations for understanding the physical processes
110 controlling the transition from shallow to deep convective clouds, some information is still missing regarding the role of water vapor and wind shear in the erosion of the nocturnal BL and the evolution to deep convection. The dry season is predominantly associated with large-scale subsidence that dries out the troposphere and suppresses large-scale deep convective clouds. However, it has been demonstrated that the probability of locally occurring daytime deep convective clouds, in the drier months (e.g., June, July, August and September), is strongly tied to both the lower-free tropospheric
115 and the BL moisture (e.g., Schiro and Neelin, 2018). In the dry season, the diurnal cycle of the convective BL includes periods of convective activity from mid-morning to afternoon that are attributed to enhanced shallow and deeper convection (Giangrande et al., 2020) in response essentially to lower free troposphere water vapor (Ghate and Kollias, 2016). This study examines the BL processes that influence convective cloud development, particularly during the dry season, when the BL contributes more to the total column moisture than during wet season (Schiro and Neelin, 2018).Line 68-79

120 3. Although the focus is different, some of the composite analysis results are actually quite very similar to Zhuang et al. 2017, eg. part of Figure 2-5. This point should be mentioned and explained specifically why presenting these similar results about shallow-deep difference are necessary for understanding the boundary layer processes. In addition, similarity, difference, and new findings regarding to the results of this study compared to Zhuang et al. 2017 should also be discussed more clearly.

125 To describe the environmental conditions in which the BL grows during the early morning, we describe vertical structure during all cloudy-BL stages, thanks to the additional radiosonde in the morning, which allowed us to investigate the rapid growth of BL. In contrast to the related study convective clouds evolution which characterizes seasonal variations (e.g., Zhuang et al., 2017), we provided an overview of environmental conditions associated with the BL evolution. We included sentences with this discussion throughout section 3.1. Also, we included a new sentence on conclusion:

130 "This finding supports the previous results (Ghate and Kollias, 2016; Zhuang et al., 2017; Tian et al., 2021), but extends the knowledge by showing in detail the different BL processes evolved in the shallow to deep convection transition. In addition, our results show that the time taken to eliminate the nocturnal stable boundary layer inversion is used to promote convection having an important effect in characterizing the convective strength in the morning transition stage." Line 559–562

135 The overall writing should be improved. There are many unclear sentences and I list some of them below as well as some minor comments. The manuscript has been completely checked by all authors. Also, we have corrected the points below:

– Line 8: I don't think the vertical wind shear presented here qualified as "intense". We rewrote the sentence: "[. . . ] well established vertical wind shear pattern." Line 8

– Line 24: convection -> convective - Corrected Line 43

140 – Line 25: central or Central? Please be consistent throughout the paper, including the title. Corrected throughout the manuscript.

– Line 79-80: This sentence should either be split or reorganized. "The vegetation cover (. . . ) nearby the intersection . . . " does not make sense. We rewrote the sentence:

The T3 was a pasture site of 2.5 km by 2 km surrounded by forest, with the forest canopy height of approximately 35m
145 (Martin et al. (2016)). Line 113-114

– Line 83: do you mean "meteorological variables at near-surface level"? Corrected Line 118

– Line 84-85: what is "conditional rain rate"? and what does "threshold" here mean for rainfall We rephrased the sentence as:

"Precipitation is used from the Present Weather Detector herein in the conditional rain rate (e.g., mean rain rates are determined only for the periods when rainfall is equal to or greater than 1 mm h$^{-1}$) to selected the days with precipitation." Line 121-122

– Line 92-94: what is the difference between three BL candidates? We rephrased this part of the manuscript according to your question and it now reads as follows:

"The BL height was retrieved by a laser ceilometer (ARM, 2014) operating at 905-nm wavelength. The ceilometer retrieves three BL height candidates provided by Vaisala BL-view software at 16 s resolution. The standard approach used consists of determining the layers associated with the backscatter gradient profile as possible BL height candidates. As aerosol concentration often occurs at the base of (or within) the entrainment zone in convective conditions or at the level of the temperature inversion capping the residual layer in neutrally stratified conditions. The first BL candidate is associated with the BL height while the others (second and third estimates) with residual layers (Poltera et al., 2017; Carneiro and Fisch, 2020). Therefore, the first BL height candidate was used to determine the height during the diurnal evolution of the atmospheric BL in 5 min average. We refer to the BL height alternatively as the BL top or the BL depth associated with the BL mixing layer. The second and third BL candidates were used to estimate the hourly average of residual layer height. [...]" Line 130-134

– Line 133-134: The ARSCL cloud top and Are these criteria need to be met simultaneously or only one is OK?

The definitions according to the height of cloud and thickness of ARSCL product and the reflectivity of the S-band radar less than 20 dBz should be satisfied simultaneously. We remove any cases from the ShCu/ShDeep analysis if deep convection occurs during the nighttime, to avoid modification in the stable stage due to precipitation processes. We rephrased the sentence:

" [...] A day was classified as representative of shallow or shallow-to-deep cloud day if it simultaneously satisfied the following day conditions" Line 176-177

– Line 150: profile -> profiles Corrected.

– Line 151: times -> time Corrected.

– Line 155-170: Why discuss temperature in between humidity variables. If there is no particular reason, I think the discussions of moisture profile should be placed together. We changed the text accordingly.

– Line 183-184: "12m/s" and "10m/s" do not matched what's shown in Figure 2d.

Thank you for noticing. There was a mistake in the label of the horizontal axis in Fig2d. Figure 2 was updated and replaced.

– Line 190-191: This sentence is hard to understand. I suggest split and rewrite to make it clearer.

We rephrased the sentence:

"The BL evolution is influenced by cooling and moistening in the first 1000 m leading to a higher RH profile throughout the lower atmosphere." Line 261-262

– Line 194: what produces a lower LCL and CIN. Please rewrite this sentence as well. We rephrased the sentence:

"Both LCL and CIN show a stronger diurnal cycle due to the strong increase in BL temperature and moisture. The environmental condition relatively moister than the environmental conditions on ShCu days leads to lower LCL and CIN on ShDeep days. ShCu days have lower free-tropospheric and BL moisture and thus higher LCL and CIN value (Zhang and Klein, 2010, 2013) creating a larger barrier for BL processes (Tian et al., 2021). Line 265- 268.

– Line 208: it is convective clouds that reduce solar radiation not precipitation. Line 208-210: This statement is not clear.

We rephrased the sentence:

" On days with ShDeep, during the cloudy mixing layer stage, clouds are deeper, with larger optical depth, reflecting more shortwave radiation. Consequently, they reduce the shortwave radiation and cool the surface. This differences is very clear when ShDeep days are compared to ShCu days during the late morning or afternoon. ShDeep days have lower net shortwave radiation than ShCu days by an average value of around 70 W m$^{-2}$ between 09:00 and 10:00 LST, and 200 W m$^{-2}$ between 12:30 and 15:00 LST." Line 284-288

– Line 220: Fig. 4c -> Fig. 4d Corrected.

– Line 241: These differences are not statistically significant due to the very large errorbar. We removed this sentence.

– Line 282: what does the plus-minus sign mean here? Is it standard deviation or confidence interval. The plus-minus means the average for the whole stable stage plus standard deviation. We rephrased the sentence:

" on average and standard deviation of $80 \pm 50$ m over the whole stable stage" Line 370-371

– Line 383: "different nighttime and environment"?

We rephrased the sentence: "During the stable stage, the combination of environment conditions and nocturnal BL characteristics favor such marked variations observed in the morning transition stage." Line 534-535

– Line 386: Please correct this sentence "2 m specific moisture, such as warm air temperature". Also, the term should be specific humidity not moisture. We changed the text accordingly.

– Figure 1e: why is there no deep convection observed? We fixed this error in Figure 1e.

– Figure 2: in the second row, x axis range should be narrowed down to better shown the difference. Figure 2 was updated and replaced.

---

## Author Comment (AC2)

Responses to referee comments on the acp-2021-87 Manuscript "Morning boundary layer conditions for shallow to deep cloud evolution during the dry season in the central Amazon", by Henkes et al.

**Referee comments 2**

We thank you for your constructive comments and suggestions to improve our study. We have carefully addresses all comments/suggestions made by reviewer. Please find below our point-by-point responses in red color, and the changes to the manuscript are in blue color response.

This manuscript used field observations to study the shallow to deep convective cloud evolution in the central Amazon. Several transition cases are comprehensively discussed. In general, this study discussed an important issue with solid analyses. However, the limited cases may not support statistically significant conclusions. Meanwhile, the significance of this study needs to be better summarized in the conclusion section. Therefore, this paper needs major revisions before potential publication in Atmos. Chem. Phys.

The main findings are shown in the conceptual model under ShCu and ShDeep conditions, which were not discussion in details in previous work:

"This finding support the previous results (Ghate and Kollias, 2016; Zhuang et al., 2017; Tian et al., 2021), but extends the knowledge by showing in detail the different BL processes evolved in the shallow to deep convection transition. In addition, our results show that the time taken to eliminate the nocturnal stable boundary layer inversion is used to promote convection having important effect in characterizing the convective strength in the morning transition stage." Line 559–562

As suggested by both referees, we included a new section with statistical analysis of BL processes, especially during stable and morning transition stages (also note our response to the specific comment below). In summary, we rewrite part of the introduction, and we include a new section with BL processes statistics (Sec. 4) and rewrite the conclusions to be more focused on the new findings.

Specific Comments:

1. The evolution of boundary layer cloud can largely differ case by case. Therefore, the mechanism and evolution pattern observed in the several cases may not be representable enough.

   Thanks for your comment. This concern was also raised by Reviewer 1. Motivated by both suggestions, a new section with BL processes statistics based on the overview of case studies is now presented in Sect. 4 (Line 460-514).

[revised manuscript text omitted]
 impacts of humidity on the development of convective clouds have been intensively investigated in previous studies (e.g., Zhang et al. 2010, 2013). The authors need to address the differences in their findings comparing to previous studies. Thank you for the suggestion. We added references to the text. Also, we added the recent GoAmazon work about diurnal cycle of clouds of Tian et al., (2021) which expands the previous classification of Zhang and Klein (2010,2013) based on US Southern Great Plains to GoAmazon main site. This new classification of ShDeep events was used to provide statistical analysis (e.g. New Sec. 4) of our conclusions which is based in the cloudy-BL stages of conceptual model.

3. As a key parameter, this study discussed the positive role of sensible heat flux on shallow-to-deep convective clouds. However, strong sensible heat also can lead to strong entertainment at the boundary layer top, which would induce dry air masses into the boundary layer. Thus, sensible also may negatively affect cloud development. Moreover, besides the impacts of sensible heat on cloud development, clouds can significantly alter sensible heat. Thus, this issue needs to be discussed more carefully. We agree. We added the information to the text about sensible heat flux on clouds and entrainment during the ShCu and ShDeep conditions:

Line 315-321 "A pronounced feature of the diurnal evolution of the sensible heat flux, often seen on ShDeep days, is the effect of sensible heat flux promoting the erosion of nocturnal BL driving a higher growth rate of the BL in the morning transition stage than on ShCu days. At the cloudy mixing layer stage, for ShDeep days, clouds affect the surface sensible heat flux, either positively by increasing the sensible heat flux due to processes associated with the arrival of the gust front or negatively by diminishing the sensible heat flux due to the transport of cool air from aloft into the BL (e.g. Oliveira et al., 2020). On the other hand, the large sensible heat flux during the cloudy mixing layer stage on ShCu days may be a response to BL turbulence and more entrainment of dry air from the free tropospheric into the BL (that drives more sensible heat flux)"

Line 345-349 [...] "While on ShCu days, the surface specific humidity is maximum around 08:00 LST followed by a drying throughout the rapid growth stage and cloudy mixing layer stage of around 2.0 g kg$^{-1}$ until the sunset, followed

by a later maximum at 19:00 LST. These findings are consistent with those of Zhang and Klein (2013), where a large sensible heat flux during the cloudy mixing layer were observed to contribute to greater entrainment of dry air into the BL in ShCu in the SGP site.

4. Some variations in the meteorological data may not cause by natural variability, but due to the instrument deficiency. In Figure 4-5, some signals seem a little bit noisy. In Figure 4f, there are some sharp decreases in sensible heat during the daytime. The authors may clarify these odd variations.

We agree that some plots of Fig. 4 and Fig. 5 in the same panel seem to have more variability. This variability is due to the different time interval resolution used in the plot, such as 1 min for the surface meteorological system and 30 min for ECOR. However, we do not believe that it is due to instrument deficiency since, for the analysis, we have used all good-quality flagged by the ARM user facility. To clarify this point, we have changed the time interval of the surface meteorological system by 30 min on average since it does not alter the results of our discussion. Thus, we have updated and replaced Figures 4-5.

---

## Referee Report (RR1)

**Review of "Morning boundary layer conditions for shallow to deep convective cloud evolution during the dry season in the Central Amazon" by A. Henkes et al. (R1)**

I appreciate the efforts of the authors to address my comments, revise the manuscript and add a new section for statistical analysis of more cases during two cases. I'm ok that the authors decide to focus on case study and use a new section of statistical analysis for more cases to support their conclusion. However, in this case, I would suggest showing all individual cases throughout the analysis and focus on discussing their overall behavior as well as case-to-case variability in Section 3 (use lighter color and thinner lines to present individual cases, as well as more solid color and thicker lines for their means), instead of just means and standard deviations, as I don't think standard deviation of merely 4 cases makes sense enough. One important point concluded by the authors is that the shorter morning transition stage of ShDeep days (or delay growth of BL in ShCu days) is due to larger sensible heat flux in ShDeep days. However, I don't think this point is convincing enough as the SH difference at this stage is actually not significant before 8 LST in Fig. 4f, nor did it supported by the additional analysis in Fig. 11. This part should be more carefully discussed. In this version of revision, I still find a lot of long and awkward sentences, grammatic error, incorrect reference to Figures and numbers. I suggest the authors should check the manuscript more carefully again and improve the English writing. Besides these, I'm overall satisfied with the response addressing my other comments. My recommendation is a major revision.

Specific comments:

Line 1: "two dry season years" → "dry seasons of two years"

Line 5: "Atmosphere" → "Atmospheric"

Line 8-9: recommend revision: "decreased duration of … is associated with …"

Line 13-16: This sentence is too long and hard to understand, please rephrase. Also, I did not recall seeing an analysis for the impact of morning transition duration on convection strength and subcloud processes.

Line 26: "others" → "other"

Line 35: "between land-atmosphere" → "between land surface and atmosphere"

Line 52: add "convection" after "shallow to deep"

Line 52-53: suggested revision: "Comparing the days with … to those with …"

Line 61-63: do you mean shallow cumulus clouds and deep convective clouds represent 22.1% and 5.2% of the total cloud cover year round, respectively, while for the dry season, the numbers are reduced to 16.6% and 1.5%, respectively? In addition, why are these numbers so small? Are these fraction of total cloud cover or cloud frequencies?

Line 66-67: not sure what this means. Are you explaining the new criteria used in Tian et al. 2021? Please rephrase.

Line 76: what is attributed to enhanced shallow and deep convection? Or do you just mean during the diurnal cycle of convective BL in the dry season, the enhanced shallow and deep convection are related to increased humidity in the lower free troposphere.

Line 84: I think you mean "adopt" not "adapt"

Line 86: I recommend to delete "until the last stage when shallow clouds evolve to deep," for clarity.

Line 130: please rephrase.

Line 131: do you mean "high aerosol concentration"?

Line 148: "allow estimating" → "estimates"

Line 150: " and distant" → "which is"

Line 159: If this stage includes (1) the sunrise, then why it "occurs two hours after sunrise"?

Line 164: should it be "It corresponds to the occurrence of a cloudy mixing layer …"?

Line 177: delete "day"

Line 180: "around 2km"? around or under?

Line 192: "former" → "latter"?

Line 193: "shallow-to-deep" → "shallow to deep"

Line 201: "allowed" → "allowed us"

Line 203: "significant number"? Do you mean "larger number"?

Line 217: "difference … important"? consider use "apparent" or "larger"

Line 265: "stronger diurnal cycle" than what? "increase in" → "diurnal cycle of"

Line 266: "relatively" → " is relatively"

Line 269: "overlap of these two zones", what zones?

Line 280: should be "diurnal evolution of convective clouds"

Line 282: add "in ShDeep days" after "cloud cover"

Line 291: do you mean "as the cloud fraction difference between them increases"?

Line 295: should be "Fig. 4c" not "Fig. 4d"

Line 298: should be "Fig. 4d" not "Fig. 4c"

Line 307: should be "Fig. 4e" not "Fig. 4c"

Line 312: "surface"? or "surface warming"?

Line 320: I don't think the sensible heat flux in morning transition stage (I suppose it's ~ 6 to 8) show significant difference between the two regimes large enough to explain the delay of BL growth in ShCu days.

Line 324: I don't see sensible heat flux on ShCu days larger than that on ShDeep days in your Fig. 4f.

Line 334: "1.1°C and 2.3°C" are not "rates", please correct the units.

Line 336: I think you mean "than ShCu days" not "unlike ShCu days"

Line 339: "noon" for "ShDeep days", 15 LST for "ShCu days"

Line 340: "drop of 3°C" during?

Line 346: " on the order of -1.0" → "by about 1.0"

Line 356: I don't think there is a square root in the equation.

Line 363: "at the time the gust front …"?

Line 374-375: I don't see a BL depth increase around 7.

Line 375 & 377: These numbers (240 and 290 W m-2) don't match with Fig. 4. In Fig. 4a, SW at 8:00 is nearly the same for two regimes, while in Fig. 4b, LW at 8:00 had a difference smaller than 20Wm-2. How can the net radiation difference be 50W m-2 here?

Line 386: "low-troposphere" → "lower troposphere"

Line 388: I think the BL depth for ShDeep days is only about 1800-1900m at 10 LST according to Fig. 6.

Line 392: "influence" → "can influence"

Line 395-396: please provide reference for this statement.

Line 403-405: sentence too long and hard to follow, please rephrase.

Line 414: where are these number (19 and 12 g kg-1)from?

Line 438: "agreement" between what?

Line 439: which is later than which by how long?

Line 445: these numbers look incorrect for me when compare to Table 2 last column, please check

Line 447: these number also look incorrect, check

Line 450-451: suggested revision: "It indicates a positive …"

Line 468: "focus" → "focuses"

Line 469: delete "significantly"

Line 472: "meteorology"? please indicate more specifically or just remove it.

Line 473: "large … small" → "larger … smaller"

Line 473: is there any particular reason to change the vertical range from 1km to 2km for IWV and VWS (Fig. 9 and 11). If not, please keep consistent.

Line 484: should be "Fig. 11d" not "Fig. 11c"

Line 488: I would suggest to test if there is a significant difference between ShCu and ShDeep for each pair of bars in Fig. 11.

Figure 8: please explain what the lowest cloud base data come from in the caption.

---

## Author Response (AR2)

Dear Editor,

We thank again the Editor and the referees comments which led to significant improvements of the quality of our manuscript. Please find below, the modifications made at the manuscript. The referees' comments are in black, our point-by-point responses are in red color, and the changes to the manuscript are in blue color.

5  Sincerely,

Alice Henkes on behalf of all coauthors

Responses to referee comments on the acp-2021-87 Manuscript "Morning boundary layer conditions for shallow to deep cloud evolution during the dry season in the central Amazon", by Henkes et al.

**Referee Comments 1**

We would like to thank referee 1 for his/her constructive comments on the manuscript, which have helped improve the presentation of our results. In what follows we present our point-by-point responses are in red color, and the changes to the manuscript are in blue color.

15  I appreciate the efforts of the authors to address my comments, revise the manuscript and add a new section for statistical analysis of more cases during two cases. I'm ok that the authors decide to focus on case study and use a new section of statistical analysis for more cases to support their conclusion. Many thanks for this comment.

1) However, in this case, I would suggest showing all individual cases throughout the analysis and focus on discussing their overall behavior as well as case-to-case variability in Section 3 (use lighter color and thinner lines to present individual cases,
20  as well as more solid color and thicker lines for their means), instead of just means and standard deviations, as I don't think standard deviation of merely 4 cases makes sense enough.

1) Following the reviewer's suggestion, we made the new version of the figures in Section 3. However, thinking about the figures, we prefer to keep the previous version (mean and one standard deviation) as the new figures did not improve significantly for a better visualization. Therefore, we copied the new figures into our response as follows on the next page.

25  2) One important point concluded by the authors is that the shorter morning transition stage of ShDeep days (or delay growth of BL in ShCu days) is due to larger sensible heat flux in ShDeep days. However, I don't think this point is convincing enough as the SH difference at this stage is actually not significant before 8 LST in Fig. 4f, nor did it supported by the additional analysis in Fig. 11. This part should be more carefully discussed.

2) The sensible heat flux values (as shown in Table 2 of the manuscript) are large and early (in the cloud-free morning) at
30  the moment of the crossover (when the sensible heat flux change from negative to positive) on ShDeep than ShCu day. For example, event 5 (20140905, ShCu) and event 7 (20140909, ShDeep) have the same time as the first positive value considering the 30 min average of ECOR sensible heat flux (of around 07 LST), but we observe 44.0 $Wm^{-2}$ on ShDeep and 0.1 $Wm^{-2}$ for ShCu day. The point is that this surface heating flux after crossover increases with time and with the amount of energy (released from the surface), causing the erosion of nocturnal BL thermal inversion and facilitating the growth of the mixed layer during
35  the morning transition stage. We rewrote the paragraph in the discussion section to clarify this:

"A pronounced feature of the diurnal evolution of the sensible heat flux, often seen on ShDeep days, is the effect of sensible heat flux promoting the erosion of nocturnal BL. After the crossover, the sensible heat flux increases with time and with the amount of energy released from the surface, driving a higher growth rate of the BL in the morning transition stage than on ShCu days.[...]"

[Figure]

**Figure 1.** Vertical profiles between 50 and 3500 m of **(a)** relative humidity, **(b)** specific humidity, **(c)** potential temperature, and **(d)** wind speed from radiosondes launched at the T3 site (from left to right) at 1:30, 7:30, 10:30, and 13:30 LST. The mean (bold lines) and the case-to-case (lighter lines) are shown for shallow convective (ShCu, red) and shallow-to-deep convective (ShDeep, blue) days.

[Figure]

**Figure 2.** Time evolution of **(a)** lifting condensation level (LCL), **(b)** convective inhibition (CIN) and **(c)** convective available potential energy (CAPE) derived from radiosondes. The mean (bold lines) and the case-to-case (lighter lines) are shown for shallow convective (ShCu, red) and shallow-to-deep convective (ShDeep, blue) days.

[Figure]

**Figure 3.** Time evolution of **(a)** net shortwave radiative flux, **(b)** net longwave radiative flux, **(c)** soil moisture at superficial layer, **(d)** soil heat flux, **(e)** latent heat flux and **(f)** sensible heat flux at the surface. The mean (bold lines) and the case-to-case (lighter lines) are shown for shallow convective (ShCu, red) and shallow-to-deep convective (ShDeep, blue) days.

[Figure]

**Figure 4.** As in Fig. 3 but for **(a)** surface air temperature, **(b)** specific humidity, **(c)** wind speed estimated by surface meteorological system, and **(d)** turbulent kinetic energy estimated by ECOR.

In this version of revision, I still find a lot of long and awkward sentences, grammatic error, incorrect reference to Figures and numbers. I suggest the authors should check the manuscript more carefully again and improve the English writing. Besides these, I'm overall satisfied with the response addressing my other comments. My recommendation is a major revision.

Line 1: "two dry season years" "dry seasons of two years" . We changed the text accordingly.

Line 5: "Atmosphere" "Atmospheric". Corrected.

Line 8-9: recommend revision: "decreased duration of … is associated with …" . We rephrased the sentence:. "Results show that the decrease in time duration of the morning transition on ShDeep days is associated with high humidity and well established vertical wind shear patterns."

Line 13-16: This sentence is too long and hard to understand, please rephrase. Also, I did not recall seeing an analysis for the impact of morning transition duration on convection strength and subcloud processes.

We rephrased the sentence:

'Under these conditions, the time duration of morning transition is used to promote convection, having an important effect on the convective BL strength evolution leading to the formation of shallow cumulus clouds and their subsequent evolution into deep convective clouds.'

Line 26: "others" "other" . Corrected.

Line 35: "between land-atmosphere" "between land surface and atmosphere". We changed the text accordingly.

Line 52: add "convection" after "shallow to deep". We added accordingly.

Line 52-53: suggested revision: "Comparing the days with … to those with …" Corrected.

Line 61-63: do you mean shallow cumulus clouds and deep convective clouds represent 22.1% and 5.2% of the total cloud cover year round, respectively, while for the dry season, the numbers are reduced to 16.6% and 1.5%, respectively? In addition, why are these numbers so small? Are these fraction of total cloud cover or cloud frequencies? Because this numbers refers to

the cloud frequencies of cloud types occurrence of seven categories (shallow, deep convection, cirrus, cirrostratus, congestus, altostratus, altocumulus ). We rephrased the sentence as:

They found that shallow cumulus clouds and deep convective clouds represent 22.1% and 5.2% of the total cloud frequencies of cloud type occurrence year round, respectively. While for the dry season, the numbers are reduced to 16.6% and 1.5%, respectively.

Line 66-67: not sure what this means. Are you explaining the new criteria used in Tian et al. 2021? Please rephrase.

We rephrased the sentence as: Their criteria associated days with local deep convective clouds to a presence or not of pre-existing or external disturbance.

Line 76: what is attributed to enhanced shallow and deep convection? Or do you just mean during the diurnal cycle of convective BL in the dry season, the enhanced shallow and deep convection are related to increased humidity in the lower free troposphere. The sentence was changed accordingly.

Line 84: I think you mean "adopt" not "adapt". Corrected

Line 86: I recommend to delete "until the last stage when shallow clouds evolve to deep," for clarity. We deleted this part of sentence.

Line 130: please rephrase. We rephrased the sentence as:

"The retrieval approach determines the layers associated with the aerosol backscatter gradient profile as possible BL height candidates."

Line 131: do you mean "high aerosol concentration"? Corrected

Line 148: "allow estimating" "estimates" Corrected.

Line 150: " and distant" "which is" Corrected.

Line 159: If this stage includes (1) the sunrise, then why it "occurs two hours after sunrise"? We rephrased the sentence as:

The complete erosion of the nocturnal BL, during dry season in Amazon region, usually occurs two hours after sunrise (Carneiro et al., 2020)

Line 164: should it be "It corresponds to the occurrence of a cloudy mixing layer ..."? The sentence was changed accordingly.

Line 177: delete "day" Done

Line 180: "around 2km"? around or under? Corrected.

Line 192: "former" "latter"? Corrected.

Line 193: "shallow-to-deep" "shallow to deep" - Changed

Line 201: "allowed" "allowed us". Corrected

Line 203: "significant number"? Do you mean "larger number"?. We changed the text accordingly.

Line 217: "difference ... important"? consider use "apparent" or "larger" . We changed "important" to "apparent" in the text.

Line 265: "stronger diurnal cycle" than what? "increase in" "diurnal cycle of" Corrected

Line 266: "relatively" " is relatively". Corrected

Line 269: "overlap of these two zones", what zones? We rephrased the sentence.

Line 280: should be "diurnal evolution of convective clouds" Corrected.

Line 282: add "in ShDeep days" after "cloud cover" Done

Line 291: do you mean "as the cloud fraction difference between them increases"? Corrected.

Line 295: should be "Fig. 4c" not "Fig. 4d". Corrected.

Line 298: should be "Fig. 4d" not "Fig. 4c" Corrected

Line 307: should be "Fig. 4e" not "Fig. 4c" Corrected

Line 312: "surface"? or "surface warming"? Corrected

Line 320: I don't think the sensible heat flux in morning transition stage (I suppose it's 6 to 8) show significant difference between the two regimes large enough to explain the delay of BL growth in ShCu days.We rephrased the sentence:

"A pronounced feature of the diurnal evolution of the sensible heat flux, often seen on ShDeep days, is the effect of sensible heat flux promoting the erosion of nocturnal BL. After the crossover, the sensible heat flux increases with time and with the amount of energy released from the surface, driving a higher growth rate of the BL in the morning transition stage than on ShCu days."

Line 324: I don't see sensible heat flux on ShCu days larger than that on ShDeep days in your Fig. 4f.
We rephrased the sentence:
'On the other hand, the more entrainment of dry air from the free tropospheric into the BL on ShCu days may be a response to BL turbulence and more sensible heat flux during the cloudy mixing layer stage.'

Line 334: "1.1°C and 2.3°C" are not "rates", please correct the units. Corrected
Line 336: I think you mean "than ShCu days" not "unlike ShCu days" Corrected
Line 339: "noon" for "ShDeep days", 15 LST for "ShCu days" We changed the text accordingly.
Line 340: "drop of 3°C" during? We rephrased the sentence:
[...] a drop of $\sim$3.0°C is observed during the cloud mixing stage due to the latent cooling from rain evaporation.

Line 346: " on the order of -1.0" "by about 1.0" Corrected
Line 356: I don't think there is a square root in the equation. Corrected
Line 363: "at the time the gust front . . ."? We corrected the sentence: "arrived at the observational site".
Line 374-375: I don't see a BL depth increase around 7. We rephrased the sentence: [...] after 7[...], there is a slight increase in the BL depth.

Line 375- 377: These numbers (240 and 290 W m-2) don't match with Fig. 4. In Fig. 4a, SW at 8:00 is nearly the same for two regimes, while in Fig. 4b, LW at 8:00 had a difference smaller than 20Wm-2. How can the net radiation difference be 50W m-2 here? These number refers to the different time of net radiation 240 W m-2 at 8:00 (ShDeep) and 290 W m-2 at around 8:45 LST (ShCu). "This stage ends around 08:00 LST, at which time the net radiation is $\sim$240 W m$^{-2}$ and the sensible heat flux is about 39 W m$^{-2}$. Moreover, on ShCu days, the BL depth grows slowly from the approximately same time and ends 0.75 h later (around 08:45 LST), at which time the net radiation is $\sim$290 W m$^{-2}$ and the sensible heat flux is about 42 W m$^{-2}$."

Line 386: "low-troposphere" "lower troposphere" Corrected
Line 388: I think the BL depth for ShDeep days is only about 1800-1900m at 10 LST according to Fig. 6. Corrected
Line 392: "influence" "can influence" . Corrected
Line 395-396: please provide reference for this statement. Done

Line 403-405: sentence too long and hard to follow, please rephrase. We removed this sentence since this information is shown in the caption of Table 2.
Line 414: where are these number (19 and 12 g kg-1) from? We added the information on the sentence as follow: [...] as reported by the soundings profile [...]
Line 438: "agreement" between what? We rephrased the sentence: [...]a good agreement of the early onset observed between SODAR and the ceilometer on ShDeep days
Line 439: which is later than which by how long?
Line 445: these numbers look incorrect for me when compare to Table 2 last column, please check We revised this numbers as suggested and it is corrected with wind speed maximum of LLJ now.
Line 450-451: suggested revision: "It indicates a positive . . ." We changed the text accordingly: It indicates a positive association of a high rate-of-change of the BL height due to the high values of IWV$_{1km}$ and VWS

Line 468: "focus" "focuses" Corrected
Line 469: delete "significantly" Done
Line 472: "meteorology"? please indicate more specifically or just remove it. We deleted the word.
Line 473: "large . . . small" "larger . . . smaller". We changed the text accordingly.

Line 473: is there any particular reason to change the vertical range from 1km to 2km for IWV and VWS (Fig. 9 and 11). If not, please keep consistent. We explore the IWV during dry season of the the two years between surface and 1km, surface and 2 km, and surface and 3 km on statistical analysis (not shown). We note that IWV, VWS are higher during the ShDeep in all layers, although the larger differences were found at layer between surface and 2 km.
" For the dry seasons of two years, higher IWV and VWS were found in 2 km than 1 km as in the case studies, reflecting the variability through the progression of the dry season."

Line 484: should be "Fig. 11d" not "Fig. 11c" Corrected
Line 488: I would suggest to test if there is a significant difference between ShCu and ShDeep for each pair of bars in Fig. 11. We included the statistically significant differences on the text (Table 1-4). However, we note that sensible heat flux do not show significant difference, since it is very sensitive to cloud cover and the time of crossover:

160    "Also, the statistical two-sided Student's *t* test showed that statistically significant differences are present for all times of IWV, VWS, longwave radiative cooling, and TKE only for at 3 LST at the 95% confidence level between ShDeep and ShCu days. The BL height during the morning transition was statistically significant at the 95% confidence level at 7LST, at the 89% confidence level for 8 LST consistent with the difference of erosion time of BL between ShDeep and ShCu days."

Figure 8: please explain what the lowest cloud base data come from in the caption. We rewrote the caption in Fig. 8:

165    [...] but for the height of the BL and lowest cloud base (black dots) from ceilometer on **(a–d)** ShCu days and **(e–h)** ShDeep days [...].

**Table 1.** Absolute values from Student's t tests for the differences between composite means of IWV (Fig. 11a) and VWS(Fig. 11b) from radiosonde vertical profiles on ShDeep and ShCu days.

|         | IVW(01:30LST) | IVW (07:30LST) | IWV (13:30LST) | VWS (01:30LST) | VWS (07:30LST) |
|---------|---------------|----------------|----------------|----------------|----------------|
| t-value | 7.12          | 7.01           | 5.50           | 3.93           | 4.30           |
| p-value | 3.16e-09      | 1.98e-08       | 1.03e-06       | 1.92e-4        | 6.79e-05       |

**Table 2.** Absolute values from Student's t tests for the differences between hourly composite means of TKE(Fig. 11c) and hourly longwave radiative(Fig. 11d) on ShDeep and ShCu days.

|         | LW (0 LST) | LW (1 LST) | LW (2 LST) | LW (3 LST) | LW (4 LST) | LW (5 LST) | LW (6 LST) |
|---------|------------|------------|------------|------------|------------|------------|------------|
| t-value | 6.23       | 6.36       | 6.16       | 5.91       | 5.99       | 5.23       | 5.96       |
| p-value | 3.16e-08   | 1.83e-08   | 4.27e-08   | 1.38e-07   | 1.43e-07   | 1.86e-06   | 1.13e-07   |

|         | TKE (0 LST) | TKE (1 LST) | TKE (2 LST) | TKE (3 LST) | TKE (4 LST) | TKE (5 LST) | TKE (6 LST) |
|---------|-------------|-------------|-------------|-------------|-------------|-------------|-------------|
| t-value | 1.38        | 1.13        | 1.81        | 2.67        | 1.34        | 0.86        | 0.25        |
| p-value | 0.16        | 0.26        | 0.07        | 0.009       | 0.18        | 0.38        | 0.79        |

**Table 3.** Absolute values from Student's t tests for the differences between hourly composite means of sensible heat flux (SH, Fig. 11e) on ShDeep and ShCu days.

|         | SH (06 LST) | SH (07 LST) |
|---------|-------------|-------------|
| t-value | -1.71       | 0.68        |
| p-value | 0.09        | 0.49        |

**Table 4.** Absolute values from Student's t tests for the differences between hourly composite means of BL Height (BLH, Fig. 11f) on ShDeep and ShCu days .

|         | BLH (0 LST) | BLH (1 LST) | BLH (2 LST) | BLH (3 LST) | BLH (4 LST) | BLH (5 LST) | BLH (6 LST) |
|---------|-------------|-------------|-------------|-------------|-------------|-------------|-------------|
| t-value | 2.36        | 0.78        | 0.64        | 0.32        | 0.52        | -0.63       | 0.93        |
| p-value | 0.02        | 0.43        | 0.52        | 0.6         | 0.52        | 0.86        | 0.35        |

|         | BLH (7 LST) | BLH (8 LST) | BLH (9 LST) |   |   |   |   |
|---------|-------------|-------------|-------------|---|---|---|---|
| t-value | 2.01        | 1.57        | 0.97        |   |   |   |   |
| p-value | 0.05        | 0.11        | 0.33        |   |   |   |   |

[Figure]

**Figure 5.** Box-and-whiskers plot of (a) IWV$_{2\ km}$ during the stable, morning transition, and cloudy mixing layer stages, (b) VWS$_{2\ km}$ at stable stage and morning transition, (c) radiative cooling during the stable stage, and (d) TKE during the stable stage, (e) sensible heat flux crossover, and (f) BL height from stable to onset of convective BL. In each box, the central bar is the median and the lower and upper limits are the first and the third quartiles, respectively. The lines extending vertically from the box indicate the spread of the distribution with the length being 1.5 times the difference between the first and the third quartiles. The red and blue boxes correspond to shallow convective (ShCu) and shallow-to-deep convective (ShDeep) days, respectively.

Responses to referee comments 2 on the acp-2021-87 Manuscript "Morning boundary layer conditions for shallow to deep cloud evolution during the dry season in the central Amazon", by Henkes et al.

**Referee Comments 2**

The authors have well addressed most of my comments. However, I still have an important concern about the mechanism behind ShDeep. Non-local factors (e.g., synoptic pattern and large-scale advection) should play critical roles in the transition. The authors should add more discussions about the separate contributions from local and non-local factors.

Thank you for this comments. We rewrote the paragraph in the discussion section to clarify this:

During the dry season, the trigger mechanism of ShDeep days is a local effect; however, it only happens when there is enough IWV, and this is controlled by non-local factors. The IWV for ShDeep is explained by the synoptic patterns as suggested by Biscaro et al. (2021). Also, Ghate and Kollias (2016) shows that large-scale moisture advection can control the relationship between local land-atmosphere interactions and diurnal precipitation. Here, we show that the enhanced water vapor at low and mid-levels is important for the characterization of the cloud-BL interaction from the stable to the rapid growth stage and triggering local deep convection in the cloudy mixing layer stage. This feature, observed during the dry season over Central Amazonia, was corroborated by Chen et al. (2020) who showed that large-scale advection is vital for characterizing land-atmosphere interactions both in magnitude and type of relationships in the transition from clear-sky to precipitation clouds over the U.S. SGP site.

**References**

Biscaro, T. S., Machado, L. A., Giangrande, S. E., and Jensen, M. P.: What drives daily precipitation over the central Amazon? Differences observed between wet and dry seasons, Atmospheric Chemistry and Physics, 21, 6735–6754, 2021.

Carneiro, R. G., Fisch, G., Borges, C. K., and Henkes, A.: Erosion of the nocturnal boundary layer in the central Amazon during the dry season, Acta Amaz., 50, 80–89, https://doi.org/10.1590/1809-4392201804453, 2020.

Chen, J., Hagos, S., Xiao, H., Fast, J. D., and Feng, Z.: Characterization of Surface Heterogeneity-Induced Convection Using Cluster Analysis, Journal of Geophysical Research: Atmospheres, 125, e2020JD032 550, 2020.

Ghate, V. P. and Kollias, P.: On the Controls of Daytime Precipitation in the Amazonian Dry Season, J. Hydrometeorol., 17, 3079–3097, https://doi.org/10.1175/JHM-D-16-0101.1, 2016.